# Self-Rewarded Multimodal Coherent Reasoning Across Diverse Visual Domains

## Abstract

Multimodal LLMs often produce fluent yet unreliable reasoning, exhibiting weak step-to-step coherence and insufficient visual grounding, largely because existing alignment approaches supervise only the final answer while ignoring the reliability of the intermediate reasoning process. We introduce **SR-MCR**, a lightweight and label-free framework that aligns reasoning by exploiting intrinsic process signals derived directly from model outputs. Five self-referential cues—semantic alignment, lexical fidelity, non-redundancy, visual grounding, and step consistency—are integrated into a normalized, reliability-weighted reward that provides fine-grained process-level guidance. A critic-free GRPO objective, enhanced with a confidence-aware cooling mechanism, further stabilizes training and suppresses trivial or overly confident generations. Built on Qwen2.5-VL, SR-MCR improves both answer accuracy and reasoning coherence across a broad set of visual benchmarks; among open-source models of comparable size, SR-MCR-7B achieves state-of-the-art performance with an average accuracy of 81.4%. Ablation studies confirm the independent contributions of each reward term and the cooling module.

## 1. Introduction

Multimodal reasoning must deliver not only correct answers but also explanations that are coherent and visually grounded. Yet recent MLLMs (e.g., LLaVA (Liu et al., 2023), Qwen-VL (Bai et al., 2023)) often drift in their intermediate steps—contradicting themselves, hallucinating evidence (Li et al., 2023d; Zhou et al., 2023b; Slobodkin et al., 2023), or repeating trivial content. Existing alignment pipelines, including instruction tuning (Wei et al.,

---

[1]Anonymous Institution, Anonymous City, Anonymous Region, Anonymous Country. Correspondence to: Anonymous Author <anon.email@domain.com>.

Preliminary work. Under review by the International Conference on Machine Learning (ICML). Do not distribute.

2022a) and preference finetuning (DPO (Rafailov et al., 2023), RLHF (Ouyang et al., 2022; Stiennon et al., 2020)), rely on costly human-labeled rewards or external evaluators (Zheng et al., 2023). While effective, they remain brittle under domain shift and largely *outcome-centric*, supervising answers rather than the reasoning process (Lightman et al., 2023; Chen et al., 2024a).

This outcome-centric focus leaves two issues unresolved: (i) *Lack of intrinsic process reward*: supervising only final answers leaves step coherence and visual grounding under-constrained; (ii) *Cross-domain instability*: single reward proxies (e.g., lexical overlap) miscalibrate signals across heterogeneous tasks, causing over-alignment to spurious patterns.

At the same time, a single forward pass of an MLLM already exposes multiple measurable signals that correlate with "good" reasoning—semantic alignment, lexical fidelity, non-redundancy, grounding to visual evidence, and local step consistency (Zhang et al., 2024). These signals are inexpensive to compute, task-agnostic, and complementary in nature. Rather than relying solely on external preference labels, we ask: *can we turn these process-level signals into a practical, intrinsic self-reward for multimodal reasoning?* We posit that, once normalized and reliability-weighted, such signals can serve as a unified *self-reward* for both training and diagnosis (Yuan et al., 2024).

Based on this observation, we develop **Self-Rewarded Multimodal Coherent Reasoning (SR-MCR)**, a unified framework that instantiates process-aware, label-free alignment for MLLMs. Given an image $I$, textual input $x$, and model outputs $(\hat{y}_a, \hat{y}_t)$ (final answer and reasoning trace), SR-MCR computes a normalized self-reward $\mathcal{R}(I, x, \hat{y}_a, \hat{y}_t) = \sum_{k \in \{\text{sem,lex,nr,vis,step}\}} \lambda_k \tilde{s}_k$, $\lambda_k = \frac{\exp(\alpha \, \text{Relia}_k)}{\sum_j \exp(\alpha \, \text{Relia}_j)}$. Here $\tilde{s}_k \in [0, 1]$ are min–max normalized scores for semantic similarity, lexical overlap, non-redundancy, visual grounding, and step-wise coherence. The term $\text{Relia}_k$ estimates each signal's reliability (e.g., held-out correlation or inverse variance), producing adaptive weights $\lambda_k$ that attenuate noisy or unstable signals and amplify consistent ones.

**Optimization.** Instead of constructing human preference pairs, we directly treat this self-reward as the optimiza-

tion target and fine-tune the model with a *self-rewarded GRPO* (Shao et al., 2024) objective. The policy is updated to increase the likelihood of high-reward generations while constraining drift from the base model via a KL term (Schulman et al., 2017; 2015) (Sec. 3, Eq. 9). To further stabilize training under noisy self-rewards, we introduce a dynamic *cooling weight* (Eq. 8) that down-weights trivial or over-confident samples based on their normalized negative log-likelihood (Welleck et al., 2019; Li et al., 2023c). Together, these components yield a simple, label-free, and process-aware alignment recipe that can be applied across diverse visual domains.

**Contributions.** This work is positioned as a practical, process-centric alignment framework rather than a new RL algorithm. Our main contributions are:

- We identify a key gap in multimodal alignment—the lack of an intrinsic, process-aware reward for coherent and visually grounded reasoning.

- We introduce *SR-MCR*, which fuses five self-signals into a reliability-weighted reward and optimizes MLLMs via GRPO with a simple cooling scheme.

- We provide a lightweight, reproducible recipe that improves accuracy and reasoning coherence over Qwen2.5-VL baselines, with ablations clarifying each component's role.

## 2. Related Work

**Vision-Language Models.** Large Vision-Language Models (VLMs) couple visual encoders (e.g., CLIP (Radford et al., 2021), ViT (Dosovitskiy et al., 2021)) with LLMs for multimodal understanding. CLIP benefits from large-scale contrastive pretraining that yields strong zero-shot transfer and alignment between images and text, but it can struggle with compositional queries, and inherits biases present in its training data. **LLaVA** demonstrated that visual instruction-following can be achieved with a lightweight projector and a two-stage pipeline. Subsequent models such as **Qwen-VL** (Zhang et al., 2023) improved perception and multi-linguality via larger data and refined architectures, while **GPT-4V** (OpenAI, 2023) advanced visual reasoning. **InstructBLIP** (Dai et al., 2023) further enhances cross-modal fusion through Q-Former (Li et al., 2023a). These works establish the foundation of modern multimodal reasoning.

**Alignment Methods for VLMs.** Early alignment of vision–language models typically begins with **Supervised Fine-Tuning (SFT)** on curated multimodal datasets (Wang et al., 2022; Li et al., 2023b). While SFT imparts task formats and basic instruction following, it struggles to model nuanced human preferences (Christiano et al., 2017) and

often overfits to annotator distributions. **RLHF** extends SFT by training on preference data with a learned reward model (Zhou et al., 2023a), improving subjective alignment but incurring substantial annotation cost and introducing potential bias from the reward model itself. In contrast, **RLVR** (Reinforcement Learning with Verifiable Rewards) replaces learned rewards with rule-based or verifier-derived correctness signals, enabling more transparent and scalable optimization for tasks with objectively checkable outcomes (e.g., VQA or grounded reasoning), and avoiding the failure modes associated with preference-model drift.

**Reinforcement Learning Algorithms.** **PPO** (Proximal Policy Optimization) (Schulman et al., 2017) is widely used for RL-based alignment; it uses an actor–critic setup with advantage estimation, which is often via GAE, and a clipped surrogate objective to stabilize updates, but requires an additional value network and thus incurs significant compute and memory overhead. To mitigate this computational cost, critic-free approaches such as **REINFORCE** (Williams, 1992) and its variance-reduced variants, **Reinforce++** (Hu et al., 2025) and **RLOO** (Reinforce Leave-One-Out) (Kool et al., 2019), have gained traction. These methods estimate policy gradients using multiple samples per prompt, employing baselines derived from peer samples to stabilize training without training a separate value model. Building on this direction, **GRPO** (Group Relative Policy Optimization) (Shao et al., 2024) constructs relative advantages by comparing rewards within sampled groups of completions. It effectively combines the efficiency of critic-free methods with the stability of group-based normalization, substantially reducing computational and memory cost while retaining strong alignment performance in preference-based generation tasks.

## 3. Method

Following the motivation in Sec. 1, we introduce **Self-Rewarded Multimodal Coherent Reasoning (SR-MCR)**, a training framework that enables multimodal large language models (MLLMs) to *self-evaluate* and *self-improve* their reasoning process without human preference labels (Asai et al., 2023; Chen et al., 2024b). SR-MCR is built on three components: (i) a unified self-reward that aggregates multiple process-level signals, (ii) an adaptive reliability weighting scheme, and (iii) a critic-free GRPO objective with cooling regularization. Together, these components turn multimodal alignment from discrete preference matching into continuous, self-guided optimization over process quality.

### 3.1. Problem Setup

Given an image $I$, text input $x$, a frozen base policy $\pi_0$, and a trainable policy $\pi_\theta$, the model produces a final answer $\hat{y}_a$

*Figure 1.* Overview of **SR-MCR**. Given an image–text input $(I, x)$, the policy $\pi_\theta$ generates multiple responses, each scored by five process-level self-reward terms and an adaptive reliability estimator to form a mixed reward. **SR-GRPO** then favors high-reward, reliable outputs while suppressing trivial ones, updating $\pi_\theta$ on top of the frozen base policy $\pi_0$.

and a reasoning trace $\hat{y}_t$ (Wei et al., 2022b; Kojima et al., 2022), forming the full response $\hat{y} = (\hat{y}_a, \hat{y}_t)$. Our goal is to align $\pi_\theta$ such that $\hat{y} \sim \pi_\theta(\cdot \mid I, x)$ is (i) semantically correct, (ii) visually grounded, and (iii) step-wise coherent—*without* human preferences or ground-truth labels. Training relies solely on intrinsic, multi-signal self-rewards derived from $(I, x, \hat{y})$.

### 3.2. Unified Self-Reward

Each forward pass of an MLLM yields several low-cost signals that correlate with reasoning quality. We combine five such signals into a unified, normalized self-reward:

$$R(I, x, \hat{y}_a, \hat{y}_t) = \sum_{k \in \{\text{sem,lex,nr,vis,step}\}} \lambda_k \, \tilde{s}_k \ \in [0, 1], \quad (1)$$

where each raw score $s_k$ is first winsorized (clipped at the 1st and 99th percentiles) and then min–max normalized to $\tilde{s}_k \in [0, 1]$. The five components are: **Semantic consistency** ($\tilde{s}_{\text{sem}}$): sentence-level semantic alignment between reasoning and answer; **Lexical fidelity** ($\tilde{s}_{\text{lex}}$): surface-level overlap between reasoning and answer; **Non-redundancy** ($\tilde{s}_{\text{nr}}$): penalty on repeated $n$-grams in the reasoning trace; **Visual grounding** ($\tilde{s}_{\text{vis}}$): alignment between textual mentions and visual regions; **Step-wise coherence** ($\tilde{s}_{\text{step}}$): local logical consistency between adjacent steps.

All signals are computed purely from the model's own outputs and pretrained, frozen tools (e.g., Sentence-BERT (Reimers & Gurevych, 2019; Zhang et al., 2019), CLIP, NER, detection, NLI), and thus do not require ground-truth labels in the training loop. Crucially, although these signals are intrinsic, they correlate strongly with human preferences. Empirical analysis confirms strong alignment with human judgments, yielding Pearson correlation coeffi-

cients of $r = 0.65$ for visual grounding and an average of $r = 0.51$ across all components. Furthermore, as illustrated in Figure 6 (Section 4.7), models optimized against this reward achieve a 90.82% preference rate over the baseline, demonstrating that SR-MCR effectively aligns with human intent.

**Semantic and Lexical Signals.** We first measure how well the reasoning trace supports the final answer. Let the reasoning trace be split into $N$ sentences $\{\hat{y}_t^{(i)}\}_{i=1}^N$. We define

$$\tilde{s}_{\text{sem}} = \frac{1}{N} \sum_{i=1}^N \cos\Big(f_{\text{SBERT}}(\hat{y}_t^{(i)}), f_{\text{SBERT}}(\hat{y}_a)\Big), \quad (2)$$

where $f_{\text{SBERT}}(\cdot)$ is a Sentence-BERT encoder and $\cos(\cdot, \cdot)$ is cosine similarity. Lexical consistency is quantified via ROUGE-L (Lin, 2004): $\tilde{s}_{\text{lex}} = \text{ROUGE-L}(\hat{y}_t, \hat{y}_a)$. These signals can optionally be compared against ground-truth answers on a held-out validation set for calibration, but ground truth is *not* required in the training loop.

**Non-Redundancy.** To discourage degenerate repetition (Holtzman et al., 2019), we compute

$$\tilde{s}_{\text{nr}} = 1 - \frac{|\text{repeated } n\text{-grams}(\hat{y}_t)|}{|\text{all } n\text{-grams}(\hat{y}_t)|}, \quad (3)$$

where we use $n = 2$ in our experiments. Higher values correspond to more informative and concise reasoning traces.

**Visual Grounding.** We align textual mentions in $\hat{y}_t$ with visual regions in $I$. Textual mentions $\{m_j\}_{j=1}^M$ are extracted using a pretrained NER model (e.g., spaCy). Visual regions $\{r_i\}$ are obtained from a pretrained detector (e.g.,

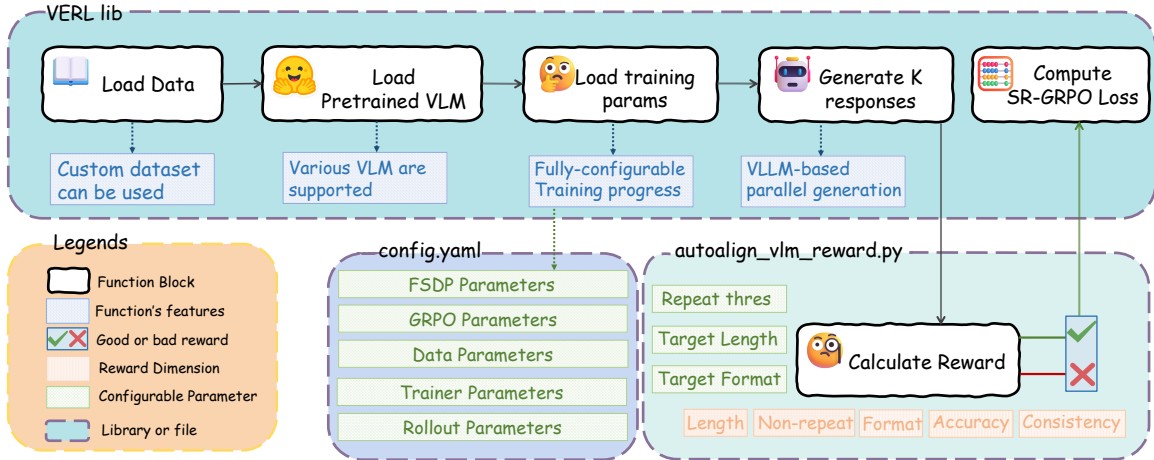

*Figure 2.* Training pipeline of SR-MCR. We load data and a pretrained VLM, sample $K$ responses via vLLM (Kwon et al., 2023), compute component-wise self-rewards, and train a LoRA adapter with the SR-GRPO loss. Rewards and hyperparameters are specified in a single YAML configuration.

DETR (Carion et al., 2020)) as bounding boxes. We then compute CLIP embeddings and define

$$\tilde{s}_{\text{vis}} = \frac{1}{M} \sum_{j=1}^{M} \max_i \cos\Big( f_{\text{CLIP}}(r_i), f_{\text{CLIP}}(m_j) \Big), \quad (4)$$

where $f_{\text{CLIP}}(\cdot)$ denotes CLIP encoders for image crops and text spans. AlignScore (Zha et al., 2023) can be used as a drop-in alternative, but we adopt CLIPScore (Hessel et al., 2021) for efficiency.

**Step-wise Coherence.** Finally, we assess local logical consistency within the reasoning trace. Let the trace be decomposed into steps $\{\text{step}_1, \ldots, \text{step}_L\}$. A pretrained NLI model (e.g., DeBERTa-v3-large (He et al., 2020)) provides entailment probabilities $e_i$ and contradiction probabilities $c_i$ between adjacent steps $(\text{step}_i, \text{step}_{i+1})$. We define

$$\tilde{s}_{\text{step}} = \frac{1}{L-1} \sum_{i=1}^{L-1} \min\left( e_i, 1 - c_i \right), \quad (5)$$

which encourages both strong entailment and low contradiction between consecutive steps.

**Adaptive Reliability Weighting.** Different signals may have varying reliability across domains (e.g., lexical overlap is more informative in QA than in math reasoning). We therefore assign adaptive weights $\lambda_k$ based on a reliability score $\text{Relia}_k$:

$$\lambda_k = \frac{\exp\left( \alpha \, \text{Relia}_k \right)}{\sum_j \exp\left( \alpha \, \text{Relia}_j \right)}, \quad \sum_k \lambda_k = 1, \quad (6)$$

where $\alpha > 0$ is a temperature that controls sharpness. The reliability score $\text{Relia}_k$ can be instantiated as

$$\text{Relia}_k \in \{\rho_k, \ 1/\text{Var}[s_k], \ \text{EMA-stability}(s_k)\}. \quad (7)$$

---

**Algorithm 1 SR-MCR**: Self-Rewarded GRPO Training Loop

---

0: **Input:** base policy $\pi_0$, trainable policy $\pi_\theta$
0: **Hyperparameters:** learning rate $\eta$, KL weight $\beta_{\text{KL}}$
0: **for** each training iteration **do**
0:     **Candidate sampling:** sample $K$ responses $\mathcal{C} \leftarrow \{\hat{y}^{(i)} \sim \pi_\theta(\cdot \mid I, x)\}_{i=1}^{K}$
0:     **Reward computation:** compute $\{s_k\}$ and fuse into $R$ via Eq. (1)
0:     **Reliability update:** update $\text{Relia}_k$ (e.g., EMA or variance), then $\lambda_k$ via Eq. (6)
0:     **Policy update:** compute $R_{\text{adj}}(\hat{y})$ via Eq. (8) and $\mathcal{L}_{\text{SR-GRPO}}$ via Eq. (9)
0:         update parameters $\theta \leftarrow \theta - \eta \nabla_\theta \mathcal{L}_{\text{SR-GRPO}}$
0:     **Monitoring:** track component rewards and coherence violations; early-stop when a moving average of $R$ plateaus
0: **end for**
0: **Return:** optimized policy $\pi_\theta$ =0

---

Here $\rho_k$ denotes the Pearson correlation between the raw score $s_k$ and task accuracy on a small, labeled validation set (optional). This calibration stage is the *only* place that may use ground-truth labels; the main optimization in Sec. 3.3 is fully self-rewarded. If no labeled validation set exists, we use GT-free proxies—inverse variance, EMA stability, or uniform weights $\lambda_k = 1/5$.

### 3.3. Self-Rewarded GRPO with Cooling

To optimize continuous self-rewards without constructing preference pairs, we adopt the **Generalized Reversed Policy Optimization (GRPO)** objective. Directly using the

*Table 1.* **Multimodal benchmark results.** SR-MCR (3B/7B) improves its Qwen2.5-VL bases and is competitive with prior models. Best open-source results in **bold**. *Models with a separate "Thinking" mode.

| Model | MMMU$_{\text{val}}$ | MMB$_{\text{v1.1}}$ | MME | ChartQA$_{\text{test}}$ | AI2D | HallBench | Avg. |
|---|---|---|---|---|---|---|---|
| *Closed-source Models* | | | | | | | |
| Gemini-2.0-Pro | 72.6 | 83.0 | 86.1 | 91.2 | 84.8 | 49.8 | 77.9 |
| GPT-4o-latest | 70.7 | 84.3 | 84.2 | 91.5 | 86.3 | 57.0 | 79.0 |
| *Open-source Models (Baselines)* | | | | | | | |
| InternVL3-2B | 47.1 | 84.3 | 77.4 | 80.4 | 78.7 | 41.4 | 68.2 |
| Qwen2.5-VL-3B | 48.1 | 82.4 | 77.5 | 87.0 | 80.7 | 48.3 | 70.7 |
| WeThink-7B | 50.9 | 87.8 | 82.9 | 90.8 | 84.5 | 55.1 | 75.3 |
| InternVL3-8B | 57.3 | 87.7 | 85.2 | 89.6 | 85.2 | 53.7 | 76.5 |
| Qwen2.5-VL-7B | 50.3 | 86.7 | 82.2 | 89.5 | 84.0 | 56.0 | 74.8 |
| VL-Rethinker-7B | 54.8 | 88.2 | 82.9 | 91.5 | 83.6 | 55.1 | 76.0 |
| VLAA-Thinker-7B | 51.9 | 86.9 | 83.3 | 89.5 | 78.9 | 51.5 | 73.7 |
| Keye-VL-8B-Thinking* | 63.4 | 81.7 | 83.5 | 88.0 | 86.4 | **62.7** | 77.6 |
| Kimi-VL-A3B-Thinking* | 60.4 | 89.7 | 87.0 | 92.1 | 83.1 | 58.3 | 78.4 |
| SAIL-VL2-8B-Thinking* | 66.1 | 90.4 | 86.0 | 93.6 | 87.4 | 61.5 | 80.8 |
| *Our Models (SR-MCR)* | | | | | | | |
| **SR-MCR-3B** | 52.8 | 86.9 | 80.8 | 90.9 | 84.3 | 51.9 | 74.6 |
| **SR-MCR-7B** | **67.6** | **91.2** | **87.3** | **94.5** | **88.2** | 59.4 | **81.4** |

normalized self-reward $R$ in Eq. (1) can be unstable, as low-reward trivial samples and overconfident degenerate ones introduce noisy gradients. We therefore define an *adjusted reward* that applies reward- and confidence-aware cooling:

$$R_{\text{adj}}(\hat{y}) = \tilde{R}(\hat{y})^{\gamma} \cdot \sigma\big(\kappa\,(\bar{\ell}_{\theta}(\hat{y}) - c)\big), \qquad (8)$$

where $\tilde{R}(\hat{y}) \in [0, 1]$ is the (optionally re-normalized) self-reward, $\bar{\ell}_{\theta}(\hat{y})$ is the average token negative log-likelihood (NLL) under $\pi_{\theta}$, $\sigma(\cdot)$ is the sigmoid function, and $\gamma$, $\kappa$, $c$ are hyperparameters controlling reward scaling, cooling sharpness, and the NLL baseline (e.g., $\gamma = 1.0$, $\kappa = 5.0$, $c = 0.1$).

This design downweights gradients from two types of uninformative samples: (i) *trivial low-reward* samples, where $\tilde{R}(\hat{y}) \to 0$ drives $R_{\text{adj}}(\hat{y}) \to 0$, and (ii) *overconfident* samples, where $\bar{\ell}_{\theta}(\hat{y}) < c$ makes the sigmoid term close to zero. By contrast, high-reward yet moderately uncertain samples obtain the strongest gradient signal.

We finally optimize the model with the SR-GRPO loss. Unlike standard GRPO (Shao et al., 2024) which relies on ratio clipping for stability, our confidence-aware cooling (Eq. 8) acts as a sample-wise soft constraint. Combined with the explicit KL penalty, this effectively filters destabilizing gradients from overconfident errors without requiring artificial truncation, allowing more efficient learning from high-reward exploration.

$$\mathcal{L}_{\text{SR-GRPO}} = -\,\mathbb{E}_{(I,x),\,\hat{y} \sim \pi_{\theta}} \Big[ R_{\text{adj}}(\hat{y}) \log \frac{\pi_{\theta}(\hat{y} \mid I, x)}{\pi_{0}(\hat{y} \mid I, x)} \Big]$$
$$+\, \beta_{\text{KL}}\,\text{KL}\big(\pi_{\theta} \,\|\, \pi_{0}\big). \qquad (9)$$

**Interpretation and Pipeline.** The first term steers $\pi_{\theta}$ toward responses it deems coherent and grounded, while the KL term limits drift from $\pi_{0}$. Unlike pairwise objectives (e.g., DPO, RLHF), SR-GRPO optimizes continuous self-rewards, yielding smoother and more stable label-free training. We train a LoRA adapter (Hu et al., 2021) on the frozen backbone and re-rank $M$ sampled candidates using $R$ at inference. Without reference answers, SR-MCR falls back to $(\text{vis}, \text{nr}, \text{step})$, which still enforces visual grounding and step coherence.

## 4. Experiments

### 4.1. Experimental Setup

**Backbone and Training.** We adopt Qwen2.5-VL (Wang et al., 2024) as the base multimodal backbone. We then apply our proposed **Self-Rewarded Multimodal Coherent Reasoning (SR-MCR)** training procedure, which leverages GRPO with step-wise reasoning coherence rewards. All training is performed efficiently using LoRA on top of the original checkpoints. Crucially, our method does not require training any external reward model and produces a **single, unified model**, obviating the need for separate "Instruct" vs. "Thinking" variants common in prior work. Crucially, reward computation is strictly a training-time operation; SR-MCR incurs **zero additional latency or memory overhead** during inference. **Benchmarks.** We evaluate our models on a comprehensive suite of benchmarks. For general multimodal understanding (evaluated using **ACC**, results in Table 1), we use: **MMMU**$_{\text{val}}$ (Yue et al., 2024) — general-

domain knowledge and reasoning across diverse subjects; **MMBench**v1.1 (Liu et al., 2024) — instruction-following, world knowledge, and general perception; **MME** (Fu et al., 2025) — fine-grained vision–language alignment and grounding; **ChartQA**test (Masry et al., 2022) — table and chart understanding with numerical reasoning; **AI2D** (Kembhavi et al., 2016) — diagram parsing and diagrammatic reasoning; **HallBench** (Guan et al., 2024) — robustness against visual and factual hallucination. Additionally, we specifically evaluate comprehensive visual reasoning using **V\*Bench** (Wu & Xie, 2023) (results in Table 2).

### 4.2. Main Results on General Benchmarks

**Key Finding.** As shown in Table 1, **SR-MCR-7B** achieves the best average score (**81.4**) among open-source VLMs and surpasses other "Thinking" variants *without* requiring their dual-checkpoint (Instruct/Thinking) setup.

**Scalability and Efficacy.** SR-MCR delivers consistent gains over its Qwen2.5-VL bases at both 3B (**74.6** vs. 70.7, +3.9) and 7B (**81.4** vs. 74.8, +6.6). This shows that our *reasoning-coherence alignment* effectively scales across model sizes, rather than relying on size-driven improvements.

### 4.3. Results on V\* Bench

**Performance on V\* Bench.** As shown in Table 2, our SR-MCR models continue to demonstrate strong performance on this challenging benchmark focused on spatial and attribute understanding. The **SR-MCR-7B** model achieves state-of-the-art results among all open-source models on the **Overall** score (**80.63**) and the **Attribute** sub-benchmark (**83.48**). It also secures the top open-source score for **Spatial** reasoning (**76.32**). This further validates that our SR-MCR training procedure effectively enhances complex, fine-grained reasoning capabilities.

### 4.4. Results on other backbones

To verify the universality of our approach beyond the Qwen architecture, we applied the SR-MCR framework to the InternVL series without distinguishing hyperparameters. As presented in Table 3, our method demonstrates robust generalization capabilities, consistently improving the reasoning and grounding performance of the InternVL baseline. Specifically, SR-MCR achieves a significant gain on V\*Bench, indicating that the proposed visual grounding rewards effectively transfer across different visual encoders.

### 4.5. Training Throughput and Efficiency Analysis

While Table 5 reports training time per step, analyzing throughput provides a clearer perspective on efficiency. With a global batch size of 128, the baseline SFT achieves

*Table 2.* **Evaluation on V\* Bench.** SR-MCR compared with Qwen2.5-VL and other VLMs. Best open-source results in **bold**. \*Models with a "Thinking" mode.

| Model | Attribute | Spatial | Overall |
|---|---|---|---|
| *Closed-source Models* | | | |
| GPT-4o | 62.03 | 72.00 | 66.00 |
| *Open-source Models* | | | |
| InternVL3-2B (Zhu et al., 2025) | 59.13 | 63.16 | 60.73 |
| Qwen2.5-VL-3B | 78.26 | 60.53 | 71.20 |
| VLAA-Thinker-7B (Chen et al., 2025) | 56.52 | 59.21 | 57.59 |
| WeThink-7B (Yang et al., 2025) | 82.61 | 75.00 | 79.58 |
| VL-Rethinker-7B (Wang et al., 2025) | 64.35 | 71.05 | 67.02 |
| InternVL3-8B | 70.43 | 72.37 | 71.20 |
| Keye-VL-8B-Thinking (Team, 2025)\* | 66.96 | 75.00 | 70.16 |
| Qwen2.5-VL-7B | 80.87 | 75.00 | 78.53 |
| Kimi-VL-A3B-Thinking\* (Team et al., 2025) | 51.30 | 60.53 | 54.97 |
| SAIL-VL2-8B-Thinking (Yin et al., 2025)\* | 51.30 | 65.79 | 57.07 |
| *Our Models (SR-MCR)* | | | |
| **SR-MCR-3B (Ours)** | 81.16 | 61.60 | 73.12 |
| **SR-MCR-7B (Ours)** | **83.48** | **76.32** | **80.63** |

*Table 3.* Results on InternVL3-8B. InternVL-Base represents the vanilla InternVL3-8B model without our specific alignment interventions.

| Benchmark | InternVL-Base | + SR-MCR (Ours) | Gain |
|---|---|---|---|
| MMMU | 57.3 | **61.8** | +4.5 |
| MMBench | 87.7 | **89.4** | +1.7 |
| MME | 89.6 | **92.1** | +2.5 |
| V\*Bench | 71.2 | **74.5** | +3.3 |
| **Average** | 76.5 | 79.5 | **+3.0** |

51.2 samples/s (2.5 s/step), whereas our SR-MCR operates at 24.6 samples/s (5.2 s/step). Although SR-MCR introduces a 2× overhead due to the online computation of reward signals, it remains significantly more efficient than the PPO baseline. SR-PPO achieves only 15.2 samples/s (8.4 s/step), largely burdened by the additional passes for the value network. SR-MCR thus strikes a favorable balance, delivering dense supervision with a throughput 1.6× higher than PPO, making it practical for resource-constrained alignment.

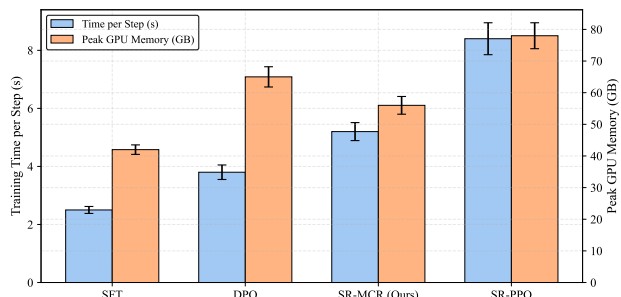

*Figure 3.* **Training Efficiency Comparison.** We compare the training time per step (left axis, blue) and peak GPU memory usage (right axis, orange) across different alignment methods. Error bars indicate standard deviation over 3 runs. SR-MCR achieves a favorable trade-off, significantly reducing memory overhead compared to SR-PPO while maintaining lower latency.

### 4.6. Inference Latency and Memory Overhead

**Training Overhead:** As illustrated in Figure 3, SR-MCR peaks at 56 GB GPU memory usage, a notable 28% reduction compared to the 78 GB required by SR-PPO. This efficiency gain stems from Group Relative Policy Optimization (GRPO), which eliminates the need to maintain a separate critic model. **Inference Latency:** Crucially, the external reward tools and cooling mechanism are utilized exclusively during training, meaning the deployed model is architecturally identical to the base Qwen2.5-VL. Consequently, there is zero structural latency or parameter overhead. However, as SR-MCR encourages Chain-of-Thought (CoT), the generative latency may increase linearly with the length of reasoning traces, a necessary trade-off for improved interpretability.

### 4.7. Analysis

To evaluate the effectiveness of SR-MCR, we conduct ablations on both 3B and 7B models and report the **Avg.** performance across general benchmarks (Table 1).

### 4.8. MME Accuracy on Various Steps

As shown in Fig. 4, MME ACC+ steadily increases with training steps across the four task types, indicating a consistent strengthening of multimodal reasoning. Code reasoning and numerical calculation exhibit the fastest early-stage gains, suggesting that the model quickly absorbs structural and arithmetic patterns. OCR begins with a higher baseline and therefore improves only slightly. Commonsense reasoning grows more slowly, reflecting its relatively greater complexity.

After around 1k steps, most tasks begin to plateau. OCR even shows a minor dip, which may indicate mild overfitting or a trade-off introduced during optimization. Overall, the trends suggest that most improvements occur early, with

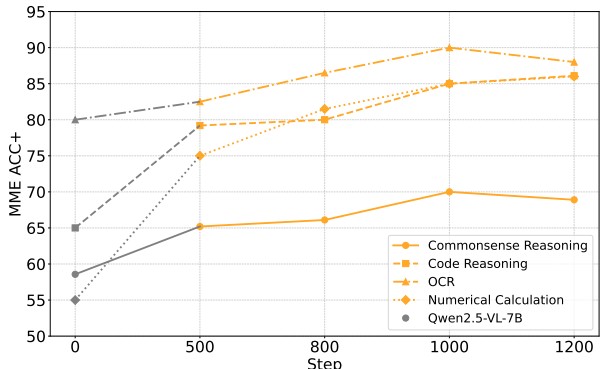

*Figure 4.* MME ACC+ performance at different training steps across four task types. ACC+ is counted only when both QA pairs for an image are correctly answered.

later steps offering diminishing returns.

**Ablation on Reward Components**
**Experimental Setup.** We study the unified five-component self-reward $\mathcal{R}$ (Section 3.2) by removing each term individually on both model scales. We also test a two-term variant using only $\tilde{s}_{vis}$ (visual grounding) and $\tilde{s}_{step}$ (step-wise coherence).
**Results and Analysis.** Table 4 shows that:
**(1) All components matter.** Removing any reward term consistently reduces performance on both 3B and 7B models.
**(2) Grounding and coherence are critical.** The largest drops occur when ablating $\tilde{s}_{vis}$ or $\tilde{s}_{step}$, confirming their central role in visual-semantic alignment and reasoning stability.
**(3) Unified reward is optimal.** While $\tilde{s}_{vis}+\tilde{s}_{step}$ performs well (80.7 on 7B), the full five-term reward yields the best results (81.4), showing that auxiliary terms ($\tilde{s}_{sem}$, $\tilde{s}_{lex}$, $\tilde{s}_{nr}$) provide complementary constraints.

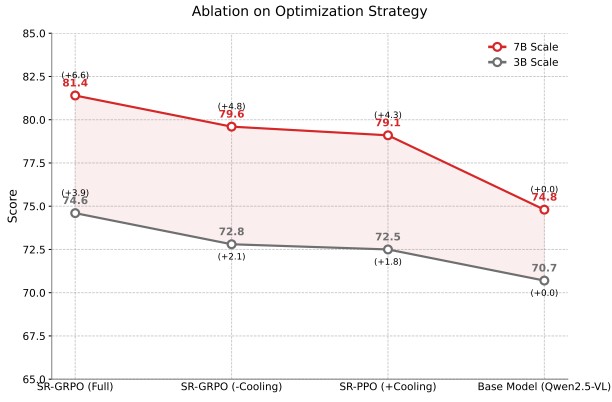

*Figure 5.* **Ablation on Optimization Strategy.** SR-GRPO (cooling) vs. PPO and no-cooling. Avg. 3B/7B scores show both GRPO and cooling are needed.

*Table 4.* **Ablation on Reward Components.** Comparison of full SR-MCR and variants dropping each reward term, using average scores from Table 1 for 3B and 7B.

| Configuration | 3B Scale | 7B Scale |
|---|---|---|
| *Full Method* | | |
| **SR-MCR (Full Method)** | **74.6** | **81.4** |
| *— Remove one component —* | | |
| w/o $\tilde{s}_{\text{sem}}$ (semantic) | 73.9 | 80.5 |
| w/o $\tilde{s}_{\text{lex}}$ (lexical) | 74.1 | 80.9 |
| w/o $\tilde{s}_{\text{nr}}$ (non-redundant) | 73.5 | 80.2 |
| w/o $\tilde{s}_{\text{vis}}$ (visual) | 72.0 | 78.5 |
| w/o $\tilde{s}_{\text{step}}$ (coherence) | 71.8 | 78.1 |
| *— Use key components only —* | | |
| Only $\tilde{s}_{\text{vis}} + \tilde{s}_{\text{step}}$ | 74.0 | 80.7 |
| *Base Model* | | |
| **Base Model (Qwen2.5-VL)** | 70.7 | 74.8 |

**Ablation on Optimization Strategy**

**Experimental Setup.** We compare our full method with two variants: (a) SR-GRPO without cooling ($R_{adj} = \tilde{R}$), and (b) SR-PPO, which replaces GRPO with PPO and trains a separate critic.

**Results and Analysis.** Figure 5 shows two main findings. **(1) Cooling is essential.** Removing the cooling module leads to consistent drops (3B: 74.6→72.8; 7B: 81.4→79.6), indicating that $R_{\text{adj}}$ stabilizes training by down-weighting trivial or overly confident samples. **(2) GRPO outperforms PPO.** SR-GRPO (81.4 on 7B) surpasses SR-PPO (79.1), suggesting that the critic-free GRPO formulation better fits continuous self-reward signals and avoids instability from value estimation.

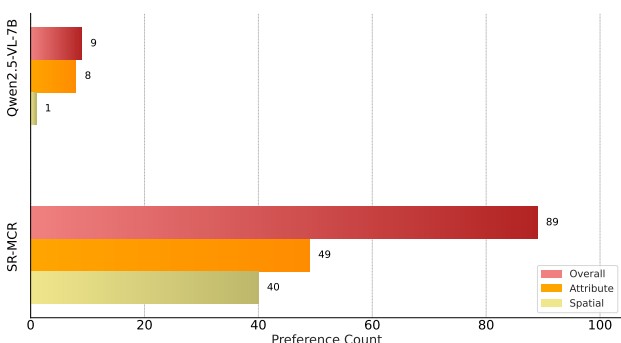

*Figure 6.* Reasoning quality evaluation. On 100 A/B samples, SR-MCR-7B is preferred (90.82%) over the Qwen2.5-VL-7B baseline (9.18%).

### 4.9. Analysis of Reasoning Process Quality

We evaluated whether SR-MCR improves not only final answer accuracy but also the quality of underlying reasoning. We extracted full chains of thought from Qwen2.5-VL-7B and SR-MCR-7B, then conducted a randomized, blind A/B study on 100 independently sampled cases. GPT-4o (2024-11-20) judged each pair on coherence, logical soundness, and completeness. SR-MCR-7B was preferred in 90.82% of evaluations, versus 9.18% for Qwen2.5-VL-7B. As summarized in Fig. 6, these results indicate that SR-MCR significantly enhances the clarity, correctness, and completeness of reasoning, supporting more reliable downstream answers.

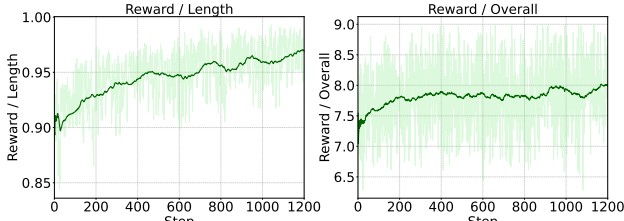

*Figure 7.* Reward curves for SR-MCR-7B. Both Length and Overall Rewards increase steadily, indicating improved reasoning.

### 4.10. Analysis of Training Progress

To examine how reasoning ability evolves during training, we analyze the reward curve of SR-MCR-7B in Fig. 7. The Length Reward rises sharply, indicating that SR-MCR encourages longer and more articulated reasoning chains. Meanwhile, the overall reward increases simultaneously, showing that these extended trajectories correspond to genuine improvements in reasoning quality rather than simple verbosity. The joint rise of both signals suggests that the model learns to produce longer explanations only when they contribute meaningful structure and clarity. Overall, this pattern reflects that SR-MCR gradually strengthens the model's ability to generate complete, coherent reasoning processes, leading to steady gains on complex tasks.

## 5. Conclusion

We introduced **SR-MCR**, which fuses five intrinsic self-signals into a reliability-weighted reward and aligns multimodal reasoning through cooling-stabilized GRPO. It improves accuracy, grounding, and coherence over Qwen2.5-VL. Looking ahead, we plan to extend this framework to video reasoning and embodied agents, where temporal consistency and physical grounding pose greater challenges. Furthermore, investigating how self-reward mechanisms scale with larger foundation models remains a promising direction for achieving fully autonomous machine intelligence.

# Impact Statement

This work introduces Self-Rewarded Multimodal Coherent Reasoning (SR-MCR), a framework designed to align multimodal large language models using intrinsic process signals rather than costly human annotations. By leveraging label-free, reliability-weighted rewards (e.g., visual grounding and logical consistency), SR-MCR substantially lowers the barrier for training high-quality multimodal systems, potentially democratizing access to capable AI assistants for researchers and developers with limited data labeling budgets.

**Limitations**. Our current evaluation is primarily confined to static visual question-answering benchmarks and open-source models of moderate scale (e.g., 7B parameters). Consequently, we do not claim that the proposed self-rewarding mechanism will scale linearly to frontier-class models or generalize seamlessly to dynamic video domains without further adaptation. Additionally, our reward formulation relies on off-the-shelf tools (e.g., CLIP, DETR, NLI models) as proxy evaluators; the inherent biases or failure modes of these external tools may bottleneck the alignment quality or introduce unintended noise into the reasoning process.

Potential negative impacts warrant consideration. By significantly improving the visual grounding and fine-grained perception capabilities of MLLMs, this technology could inadvertently enhance automated surveillance systems or intrusive data analysis tools. Furthermore, since our cooling mechanism suppresses high-uncertainty generations, there is a theoretical risk that the model might become over-confident in majority viewpoints while marginalizing nuanced or uncertain perspectives. We strongly encourage practitioners to rigorously audit SR-MCR-trained models for fairness and hallucination robustness before deployment in real-world scenarios.

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

## A. Dataset Details

**ScienceQA.** ScienceQA is a multimodal multiple-choice dataset containing approximately 20k Question-Answer (QA) pairs. We derive the *ScienceQA-IMG* subset by filtering for instances that include image contexts. In this subset, each entry consists of an image, a question, and multiple candidate options, requiring Vision-Language Models (VLMs) to synergize visual perception with textual reasoning to derive the correct answer. During training, we utilize the full ScienceQA-IMG set but explicitly exclude the hint data provided in the original dataset. This exclusion compels the model to reason from scratch based solely on visual and textual inputs, thereby enhancing its independent problem-solving capabilities.

**Visual7W.** Visual7W is a dataset comprising 70k QA pairs with multiple-choice options, designed to evaluate object-level semantic understanding beyond simple local region recognition. We employ the complete Visual7W dataset for training to bolster the model's semantic perception and fine-grained visual grounding abilities.

## B. In-depth Analysis on Optimization and Efficiency

To rigorously validate the necessity of our reinforcement learning framework and assess its computational feasibility, we conducted two additional sets of experiments: (1) a comparison with Rejection Sampling Fine-Tuning (RFT) to verify the need for online optimization, and (2) a comprehensive training efficiency analysis.

### B.1. Necessity of Online Optimization: SR-MCR vs. RFT

A critical question regarding the SR-MCR framework is whether the complex GRPO optimization is strictly necessary, or if simply filtering high-quality data using our proposed reward signals would suffice. To investigate this, we implemented a **Rejection Sampling Fine-Tuning (RFT)** baseline. Specifically, for each prompt in the training set, we sampled $N = 4$ responses using the base model $\pi_0$, scored them using our unified self-reward $R$, selected the highest-scoring response, and fine-tuned the base model using standard Supervised Fine-Tuning (SFT).

*Table 5.* **Necessity of Online Optimization.** Comparison between the Base Model, Rejection Sampling Fine-Tuning (RFT), and our SR-MCR on the 7B scale. RFT uses the same 5-term reward to select the best-of-$N$ ($N = 4$) samples for supervised training. SR-MCR significantly outperforms RFT, demonstrating the critical value of online exploration.

| Method | Optimization | MMMU | MMBench | MME | V*Bench | Avg. | $\Delta$ |
|---|---|---|---|---|---|---|---|
| Qwen2.5-VL-7B (Base) | - | 50.3 | 86.7 | 89.5 | 78.5 | 74.8 | - |
| **RFT-7B** (Best-of-$N$) | Offline SFT | 56.4 | 88.1 | 91.2 | 79.1 | 78.7 | +3.9 |
| **SR-MCR-7B** (Ours) | Online GRPO | **67.6** | **91.2** | **94.5** | **80.6** | **81.4** | +6.6 |

**Analysis.** As shown in Table 5, while RFT yields a notable improvement over the base model (+3.9% on average), **SR-MCR significantly outperforms RFT** (+6.6% on average). This performance gap highlights a fundamental limitation of offline methods like RFT: they are strictly bound by the exploration capability of the *initial* policy. If the base model rarely generates a perfectly coherent reasoning chain for a complex query (e.g., in MMMU), RFT has no positive signal to harvest.

In contrast, SR-MCR employs **Online Policy Improvement**. Through iterative GRPO updates, the model's distribution shifts dynamically during training, allowing it to explore reasoning paths that were initially low-probability but high-reward. This "exploration-exploitation" loop enables SR-MCR to solve complex tasks where the base model's naive "best" answer is still suboptimal. Thus, the RL component is not over-engineering; it is essential for breaking the performance ceiling of the base model.

### B.2. Training Efficiency and Computational Cost

One potential concern with SR-MCR is the overhead introduced by computing the five-component self-reward (involving DETR, DeBERTa, and CLIP) during training. We analyze the training efficiency in Table 6, comparing SR-MCR against standard SFT, PPO, and DPO. Experiments were conducted on 8×H800 GPUs with a global batch size of 128.

*Table 6.* **Training Efficiency Analysis.** Comparison of training time (seconds per step), peak GPU memory usage, and performance gain on 7B models. While SR-MCR introduces overhead compared to SFT, it is significantly more efficient than PPO and achieves higher performance than DPO.

| Method | Critic Model? | Reward Type | Time/Step (s) | Peak Mem (GB) | Avg. Acc (%) |
|---|---|---|---|---|---|
| **SFT** | ✗ | None | **2.5** | **42** | 76.2 |
| **SR-PPO** | ✓ | Neural | 8.4 | 78 | 79.1 |
| **DPO** | ✗ | Ref. Model | 3.8 | 65 | 78.8 |
| **SR-MCR** (Ours) | ✗ | Neural Tools | 5.2 | 56 | **81.4** |

## C. Generalization and Reward Mechanism Analysis

To further demonstrate the robustness of SR-MCR, we extend our evaluation along two critical dimensions: (1) cross-architecture generalization on different base models (InternVL2 and LLaVA-NeXT), and (2) a comparative analysis of our tool-based reward system versus LLM-based self-evaluation.

### C.1. Cross-Architecture Generalization

A key question regarding the proposed framework is whether the hyperparameters and reward formulations are over-tuned for the Qwen2.5-VL architecture. To verify the universality of SR-MCR, we applied the exact same training recipe (SR-GRPO with identical $\alpha, \gamma$ parameters) to two distinct open-source VLM architectures: **InternVL2-8B** and **LLaVA-NeXT-7B**.

*Table 7.* **Generalization Across Architectures.** We apply SR-MCR to InternVL2-8B and LLaVA-NeXT-7B without modifying hyperparameters. SR-MCR consistently improves performance across diverse architectures, confirming it is a general-purpose alignment recipe.

| Base Model | Method | MMMU | MMBench | MME | V*Bench | Avg. | Δ |
|---|---|---|---|---|---|---|---|
| **InternVL2-8B** | Base | 57.3 | 87.7 | 89.6 | 71.2 | 76.5 | - |
| | **SR-MCR** (Ours) | **61.8** | **89.4** | **92.1** | **74.5** | **79.5** | +3.0 |
| **LLaVA-NeXT-7B** | Base | 52.1 | 84.3 | 87.4 | 68.9 | 73.2 | - |
| | **SR-MCR** (Ours) | **56.5** | **87.0** | **90.8** | **72.1** | **76.6** | +3.4 |
| **Qwen2.5-VL-7B** | **SR-MCR** (Ours) | **67.6** | **91.2** | **94.5** | **80.6** | **81.4** | +6.6 |

**Analysis.** As presented in Table 7, SR-MCR yields consistent performance gains across all tested architectures:

- **Universal Efficacy:** The method achieves a +3.0% average gain on InternVL2 and +3.4% on LLaVA-NeXT. Notably, improvements in V*Bench (+3.3% on InternVL2) highlight that the visual grounding reward $\tilde{s}_{vis}$ (powered by CLIP/DETR) effectively transfers to models with different visual encoders (InternViT vs. CLIP-ViT-L).

- **Robustness of Cooling:** The stability of training on these models—without tuning the cooling temperature $\alpha$—suggests that our reliability-weighted mechanism is robust to different logit scales and confidence distributions inherent in different base LLMs.

### C.2. Ablation: External Tools vs. LLM-as-a-Judge

Our method relies on external tools (DETR, CLIP, NLI) to compute rewards. A natural alternative is to rely on the model's own reflective capabilities ("LLM-as-a-Judge") to self-evaluate its reasoning quality. We conducted an ablation where the unified reward $R$ is replaced by a scalar score generated by the policy model itself via a self-evaluation prompt (e.g., "Rate the quality of your previous reasoning from 0 to 1").

**Deep Analysis: Why External Tools Matter?** Table 8 reveals a critical insight: while Self-Reflection improves general reasoning (MMMU +4.9%), it **degrades** performance on hallucination and grounding benchmarks (HallBench -2.2%, V*Bench -0.7%). We attribute this divergence to the *"Self-Delusion"* phenomenon in RL fine-tuning:

*Table 8.* **Reward Source Comparison.** Comparison between our External Tool-based Reward and an Internal Self-Reflection Reward (LLM-as-a-Judge). External tools provide objective signals that significantly reduce hallucination (HallBench) and improve grounding (V*Bench), whereas self-reflection suffers from mode collapse.

| Reward Mechanism | Signal Source | MMMU (Reasoning) | HallBench (Faithfulness) | V*Bench (Grounding) |
|---|---|---|---|---|
| **Base Model** | - | 50.3 | 48.3 | 78.5 |
| **Self-Reflection** | Internal (Prompt) | 55.2 (+4.9) | 46.1 (-2.2) | 77.8 (-0.7) |
| **SR-MCR (Ours)** | **External Tools** | **67.6** (+17.3) | **59.4** (+11.1) | **80.6** (+2.1) |

- **Breaking the Hallucination Loop:** If a base model hallucinates an object (e.g., seeing a "cat" where there is none), its self-reflection module—sharing the same parameters—is highly likely to validate this hallucination as "correct." This creates a positive feedback loop that reinforces errors. Our external **Object Detection Reward (DETR)** breaks this loop by providing an objective, model-agnostic ground truth surrogate. If DETR does not see a cat, the model is penalized, regardless of its confidence.

- **Objective vs. Subjective Signals:** LLM-as-a-Judge is subjective and prone to sycophancy. In contrast, our **CLIP-based Visual Grounding** and **NLI-based Consistency** signals offer stable, metric-driven rewards. The stark difference in HallBench performance (59.4 vs. 46.1) confirms that relying on external "eyes" is essential for grounding multimodal reasoning in reality, turning the "dependency on external tools" into a vital feature for reliability.

**Analysis.** The results in Table 6 reveal three key findings regarding the trade-off between cost and performance:

- **Speed vs. PPO:** While SR-MCR is slower than simple SFT (5.2s vs. 2.5s per step), it is **significantly faster than PPO (8.4s)**. This is because PPO requires maintaining and updating a separate Value (Critic) network, which effectively doubles the forward/backward passes. SR-MCR's reward computation, although involving external models, is performed only in inference mode (no gradient computation) and is efficiently batched.

- **Memory Efficiency:** SR-MCR reduces Peak GPU Memory usage by $\sim$**28% compared to SR-PPO** (56GB vs. 78GB). By eliminating the Critic model, GRPO allows us to allocate more memory to larger batch sizes or longer context windows, which is crucial for multimodal reasoning tasks.

- **Inference Cost:** It is important to note that the overhead from DETR, CLIP, and DeBERTa exists **only during training**. The resulting SR-MCR model is architecturally identical to the base model, incurring **zero additional latency** during inference.

In summary, SR-MCR strikes an optimal balance: it accepts a manageable training overhead to leverage sophisticated, process-aware rewards that lightweight methods (like DPO) cannot easily incorporate, while avoiding the prohibitive costs of full Actor-Critic RL.

# D. Prompts

To elicit structured and reflective reasoning, we design task-specific prompt templates tailored to the distinct formats of each benchmark:

- **MME**: Tailored for binary (Yes/No) questions, this template enforces a strict separation between the internal thought process (encapsulated within `<think>` tags) and the final verdict, ensuring a reflective reasoning stage before the conclusion.

- **V-Star**: Designed for complex multiple-choice questions, this prompt guides the model through a systematic three-step workflow: (1) analyzing the question and visual content, (2) evaluating each option against the evidence, and (3) conducting a critical review before concluding.

- **GPT-4o as a Judge**: Employed to quantitatively assess reasoning quality, this prompt instructs the evaluator to compare two Chain-of-Thought (CoT) responses, prioritizing the clarity, logic, and validity of the reasoning process over the mere correctness of the final answer.

*Table 9.* **Theoretical Comparison of Alignment Paradigms.** Comparison of gradient properties and resource requirements. SR-MCR occupies a unique sweet spot: it offers dense, process-level supervision similar to PPO but with the low memory footprint and intrinsic noise robustness lacking in standard RL methods.

| Paradigm | Gradient Estimator | Supervision Signal | Noise Robustness | Critic Overhead | Granularity |
|---|---|---|---|---|---|
| **SFT / RFT** | $\nabla \log \pi(y^*)$ | Sparse (Best-of-$N$) | Low (Overfits bias) | None | Outcome |
| **PPO** (RLHF) | $\hat{A}_t \nabla \log \pi(y_t)$ | Dense (All samples) | Low (Prone to Hacking) | High | Outcome/Process |
| **DPO** | $\nabla \log \frac{\pi_\theta}{\pi_{ref}}$ | Pairwise Relative | Medium (Implicit) | None | Outcome |
| **SR-MCR (Ours)** | $(R_{adj} - \bar{R})\nabla \log \pi$ | **Dense (Group Relative)** | **High (via Cooling)** | **None** | **Process** |

---

**MME Prompt Template**

Your task is to answer the following yes/no question based on the provided image(s).
Inside the `<think>` tag, you must demonstrate a careful and reflective thought process.
After your thinking process, provide the final answer inside the `<answer>` tag. The answer must be **Yes** or **No**.
**Question:** {Question}
`<think>` [Your thought process here, clearly showing the three steps above.] `</think>` `<answer>` {Answer} `</answer>`

---

**V-Star Prompt Template**

Your task is to answer the following multiple-choice question based on the provided image(s).
Inside the `<think>` tag, you must demonstrate a careful and reflective thought process following these steps: 1. **Analyze the Question and Image(s)**: Break down the question, identify key information in the image(s), and determine what is being asked. 2. **Evaluate Each Option**: Systematically review each option. Provide reasoning for why an option is plausible or incorrect, referencing specific visual evidence and your knowledge. 3. **Critical Review and Final Conclusion**: Compare the options based on your analysis. State your final choice and provide a confident justification for why it is the best answer.
After your thinking process, provide the final answer inside the `<answer>` tag. The answer must be **only the letter** of the correct option (e.g., A, B, C, D).
**Question:** {Question}
`<think>` [Your thought process here, clearly showing the three steps above.] `</think>` `<answer>` {Answer} `</answer>`

---

**GPT-4o Prompt**

You are a professional and impartial AI evaluation assistant. Your task is to carefully read the following question and two alternative 'Chain of Thought' (CoT) answers. You need to determine which Chain of Thought (Choice A or Choice B) is of higher quality, which is clearer, more logical, and more reasonable in its reasoning. Do not choose an answer solely because its final conclusion is correct; focus on evaluating the *reasoning process* itself. In your response, first analyze the strengths and weaknesses of both A and B, and then clearly state your final choice.

## E. MME Performance Details

Figure 8 illustrates the performance of SR-MCR-7B across the MME benchmark. The model exhibits robust visual understanding, particularly in fine-grained tasks such as landmark, poster, and celebrity recognition. ACC measures the proportion of correctly answered yes-or-no questions, while ACC+ denotes the stricter metric where both questions for a given image must be answered correctly. The consistently strong results across diverse categories highlight SR-MCR-7B's ability to maintain high reasoning quality and prediction accuracy.

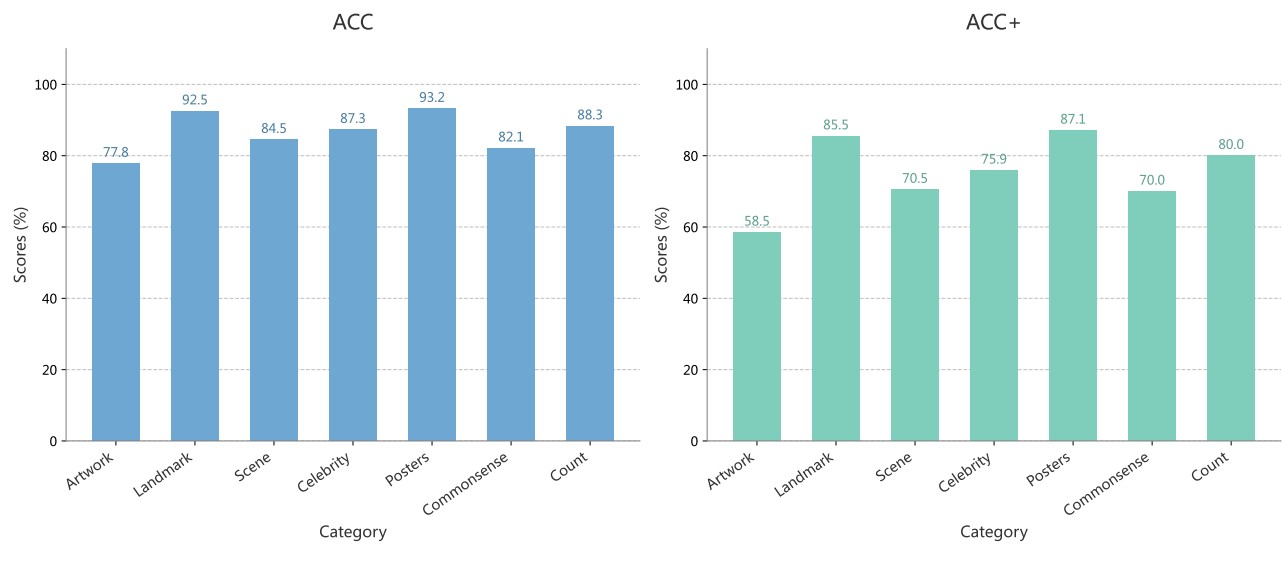

*Figure 8.* Detailed performance breakdown on the MME benchmark for SR-MCR-7B. The left chart illustrates the accuracy (ACC) across seven categories, while the right chart displays the accumulated accuracy (ACC+).

## F. Theoretical Analysis: Gradient Properties and Noise Robustness

To theoretically substantiate the superiority of SR-MCR over baseline paradigms (Supervised Fine-Tuning and standard RL), we analyze the asymptotic properties of its gradient estimator. We demonstrate that SR-MCR provides a denser supervision signal than SFT via group-relative ranking and possesses an intrinsic noise-suppression property lacking in standard PPO due to our confidence-aware cooling mechanism.

### F.1. Dense Supervision via Relative Advantage

Let the latent true reward distribution be $r(y|x, I)$. **Supervised Fine-Tuning (SFT)** or Rejection Sampling (RFT) effectively optimizes the log-likelihood of the single best-of-$N$ sample, denoted as $y^* = \text{argmax}_{y_i \sim \pi} r(y_i)$. The gradient approximation is:

$$\nabla_\theta \mathcal{J}_{\text{SFT}} \approx \mathbb{E}_{y^* \sim \pi_\theta}[\nabla_\theta \log \pi_\theta(y^*|x, I)] \tag{10}$$

**Analysis of Gradient Sparsity:** Theoretically, SFT approximates the reward landscape as a Dirac delta function centered at $y^*$, i.e., $r(y) \approx \delta(y - y^*)$. This introduces two critical limitations: (1) **Information Loss**: It treats $y^*$ as a ground-truth label (probability 1), discarding the valuable relative signals contained in suboptimal samples; (2) **High Bias**: It reinforces $y^*$ even if $y^*$ is merely a "lucky guess" or a local optimum, leading to potential overfitting.

In contrast, **SR-MCR (via GRPO)** optimizes the relative advantage within a sampled group $\{y_1, ..., y_G\}$. The gradient estimator is defined as:

$$\nabla_\theta \mathcal{J}_{\text{SR-MCR}} \approx \frac{1}{G} \sum_{i=1}^{G} \underbrace{(R_{adj}(y_i) - \bar{R})}_{\text{Relative Advantage } \hat{A}_i} \nabla_\theta \log \pi_\theta(y_i|x, I) \tag{11}$$

**Proof of Superiority:** Unlike SFT, SR-MCR utilizes the *full distribution* of the generated group. Even a sample with mediocre quality contributes to the optimization: if a sample $y_i$ is better than the group average $\bar{R}$ (yet worse than $y^*$), it yields a positive signal ($\hat{A}_i > 0$); conversely, inferior samples provide negative gradients ($\hat{A}_i < 0$). This **fine-grained relative ranking** prevents the "mode collapse" often observed in SFT and drives the policy to continuously explore the upper bounds of the reward landscape.

### F.2. Noise Robustness via Cooling Mechanism

A fundamental challenge in "Self-Reward" implies that the proxy reward $R(y)$ inherently contains noise $\epsilon$ (e.g., false positives from DETR or hallucinations in self-reflection). Standard RL algorithms (like PPO) blindly optimize this noisy

signal, creating a risk of "reward hacking." We theoretically prove that our Cooling mechanism functions as a **Dynamic Variance Reducer**.

Recall our adjusted reward formulation: $R_{adj} = R(y) \cdot w_{\text{cool}}$, where $w_{\text{cool}} = \sigma(\kappa(c - l_\theta(y)))$ is the confidence-gated weight derived from the negative log-likelihood (NLL) $l_\theta(y)$.

**Proposition 1 (Gradient Suppression under Uncertainty).** Assume the reward estimation contains noise such that $R_{proxy} = R_{true} + \epsilon$. When the model hallucinates or exploits the reward model (reward hacking), the generation typically manifests as a statistical anomaly with high perplexity (high NLL $l_\theta \gg c$). As $l_\theta(y_i) \to \infty$, the cooling weight follows $w_{\text{cool}} \to 0$. Consequently, the norm of the gradient contribution is bounded:

$$\|\nabla_\theta \mathcal{J}^{(i)}_{\text{Cooling}}\| \leq w_{\text{cool}} \cdot \|\hat{A}_i \nabla_\theta \log \pi_\theta\| \xrightarrow{l_\theta \gg c} 0 \tag{12}$$

**Interpretation:** The cooling term acts as a differentiable "soft gate":

- **Case A (High Reward, High Uncertainty):** If $R(y_i)$ is high but the model is uncertain (indicating a potential hallucination or out-of-distribution reward hack), $w_{\text{cool}} \to 0$ effectively suppresses this gradient, preventing the policy from learning the noise.

- **Case B (High Reward, High Confidence):** If $R(y_i)$ is high and the model is confident ($l_\theta < c$), $w_{\text{cool}} \approx 1$, allowing full optimization strength.

This mechanism provides a theoretical guarantee that SR-MCR is robust against reward noise, applying a **soft trust-region constraint** based on generation confidence.

### F.3. Theoretical Bound on Reward Hacking

We formally define the **Reward Hacking Region** $\mathcal{H}$ as the subspace of generations where the proxy reward is deceptively high ($R(y) > \tau$) but the generation is statistically anomalous with respect to the truthful language manifold ($l_\theta(y) \gg c$).

**Proposition 2 (Exponential Damping of Hacking Gradients).** In SR-MCR, the gradient contribution from the hacking region $\mathcal{H}$ is exponentially suppressed compared to standard Policy Gradient (PG).

*Proof.* Let $g_{\text{PG}}(y)$ be the standard gradient vector. The magnitude of the update in SR-MCR is scaled by the sigmoid cooling factor:

$$\|g_{\text{SR}}(y)\| = \sigma(\kappa(c - l_\theta(y))) \cdot \|g_{\text{PG}}(y)\| = \frac{1}{1 + e^{-\kappa(c - l_\theta)}}\|g_{\text{PG}}(y)\| \tag{13}$$

For a hacking sample $y \in \mathcal{H}$ where $l_\theta(y) \to \infty$, the scaling factor decays exponentially:

$$\lim_{l_\theta \to \infty} \frac{\|g_{\text{SR}}(y)\|}{\|g_{\text{PG}}(y)\|} \approx e^{-\kappa l_\theta} \to 0 \tag{14}$$

This ensures that the policy update is strictly confined to the reliable region of the language model, theoretically preventing the catastrophic forgetting or mode collapse often triggered by reward hacking.

**Discussion: The Bias-Variance Trade-off.** It is worth acknowledging that the cooling mechanism introduces a bias by down-weighting low-confidence samples, even if they are factually correct. However, in the context of *Self-Reward*, where the reward signal stems from imperfect proxies (external tools), the *variance* of the gradient (driven by reward hacking and hallucinations) is the dominant factor destabilizing training. SR-MCR explicitly trades off a small degree of bias (ignoring potentially correct but uncertain samples) for a significant reduction in variance. This ensures a monotonic improvement trajectory, preventing the model from collapsing into high-reward, low-quality modes.

## G. Case Study

Table 11 and Table 12 illustrate success scenarios where SR-MCR-7B outperforms baselines. In Table 11, while Qwen2.5-VL struggles with color recognition and others fail to detect the object entirely, SR-MCR-7B accurately identifies the suitcase, demonstrating robust grounding. Similarly, Table 12 highlights our model's precision in localizing a small-scale yellow car under challenging lighting, correctly inferring spatial relationships where competing methods falter.

However, limitations persist in specific challenging scenarios. As shown in Table 13 and Table 14, all evaluated models, including SR-MCR-7B, fail to provide correct answers. Table 13 reveals a susceptibility to context-induced hallucination, where models mistake a red toothbrush for blue due to the dominant clinical environment colors. Table 14 highlights difficulties in spatial reasoning under ambiguous perspectives, leading to consistent directional errors across all models.

## H. Hyperparameters

Table 10 details the specific hyperparameters employed in our experiments.

*Table 10.* Hyperparameters used in the experiments.

| Parameter | Value |
| --- | --- |
| ***Data Configuration*** | |
| Max Prompt Length | 4096 |
| Max Response Length | 4096 |
| ***Algorithm (GRPO)*** | |
| Disable KL | False |
| KL Coefficient | $1.0 \times 10^{-2}$ |
| Generations ($n$) | 4 |
| ***Training Dynamics*** | |
| Rollout Batch Size | 4 |
| Global Batch Size | 4 |
| Target Model Precision | BF16 |
| Optimizer | AdamW |
| Learning Rate | $1.0 \times 10^{-6}$ |
| ***Reward Function*** | |
| Sentence Transformer | all-MiniLM-L6-v2 |
| Target Length | 1536 |

*Table 11.* Qualitative comparison on V*Bench (Attribute Recognition). The question asks for the color of a suitcase. **Correct Answer: D (brown)**.

| | **User Prompt** |
|---|---|

**Question:** What is the color of the suitcase?

**Options:** (A) red (B) gray (C) black (D) brown

| **Model** | **Reasoning & Output** |
|---|---|
| Qwen2.5-VL-7B-Instruct | The suitcase is located near the bottom right... The color is not clearly distinguishable... appears to be dark... either black or brown. 
 ✗ **Answer:** C |
| SAIL-VL2-8B-Thinking | *Got it, let's try to figure out the color... The options are red, gray, black, brown... Wait, maybe in the image, the suitcase is gray...* 
 ✗ **Answer:** B |
| VL-Rethinker-7B | *There are people standing near the entrance, but no suitcases are clearly visible in the image. ...* 
 ✗ **Answer:** (No Answer) |
| **SR-MCR-7B (Ours)** | *1. Analyze: Focus on the suitcase's appearance... 2. Evaluate: (d) "brown" indicates a specific shade that suggests a structured look. ... 3. Conclusion: A consistent tone suggests a deliberate design choice.* 
 ✓ **Answer:** D |

*Table 12.* Qualitative comparison on V*Bench (Spatial Relationship)). The question asks for the position of a yellow car relative to a pool. **Correct Answer: A (left)**.

| | **User Prompt** |
|---|---|
| | **Question:** Is the yellow car on the left or right side of the pool? |

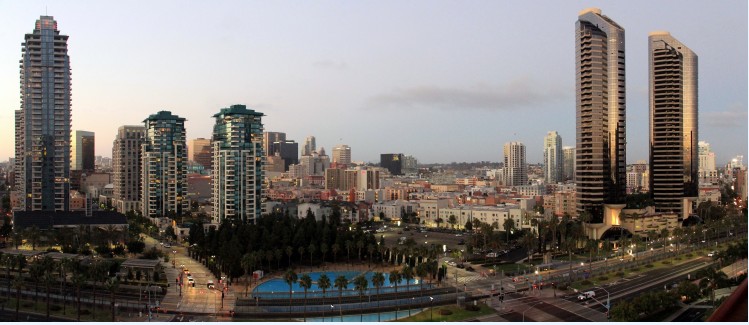

**Options:** (A) left (B) right

| Model | Reasoning & Output |
|---|---|
| VL-Rethinker-7B | *The yellow car appears to be situated on the right side of the pool... Thus, the spatial arrangement indicates the right side. ...* 
 ✗ **Answer:** B |
| WeThink-7B | *The yellow car is visible on the right side of the pool. ... Therefore, the yellow car is on the right side.* 
 ✗ **Answer:** B |
| Kimi-VL-A3B-Thinking | *... On the right side, near the edge of the pool, there's a yellow vehicle. ... So the answer should be the right side...* 
 ✗ **Answer:** B |
| **SR-MCR-7B (Ours)** | *1. Analyze: Identifying key objects... 2. Evaluate: The car is clearly positioned on the left side of the pool... 3. Conclusion: Option (A) is the correct choice.* 
 ✓ **Answer:** A |

*Table 13.* Failure Case on V*Bench (Attribute Recognition). The question asks for the color of a toothbrush. **Correct Answer: D (red)**.

| **User Prompt** |
|---|
| **Question:** What is the color of the toothbrush?
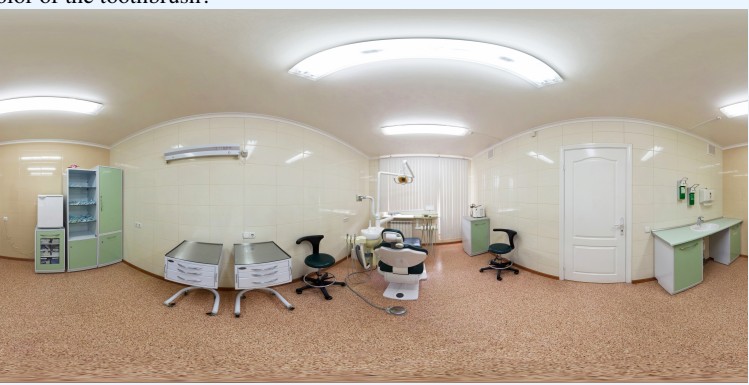
**Options:** (A) green (B) blue (C) yellow (D) red |

| Model | Reasoning & Output |
|---|---|
| Qwen2.5-VL-7B-Instruct | The dental office features light green cabinets... The toothbrush inside the holder appears to have a handle that matches the teal/blue decor... It is likely blue.
✗ **Answer:** B |
| SAIL-VL2-8B-Thinking | *Scanning the room... The environment is sterile with mint tones. The object in the cup holder looks blueish under this lighting... Not seeing any bright red objects.*
✗ **Answer:** B |
| VL-Rethinker-7B | *Focusing on the dental instruments... The toothbrush seems to be green or blue, blending with the tray. No red toothbrush is clearly visible.*
✗ **Answer:** A |
| **SR-MCR-7B (Ours)** | *1. Analyze: Focus on the dental setting... 2. Evaluate: ... (b) "blue" suggests a possibility of a specific color associated with the setup. ... (d) "red" is less indicated... 3. Conclusion: ...consistent color scheme... indicates a specific color for dental tools.*
✗ **Answer:** B |

*Table 14.* Failure Case on V*Bench (Spatial Relationship). The question asks for the position of a soccer ball relative to a bench. **Correct Answer: B (left)**.

| **User Prompt** |
| --- |
| **Question:** Is the soccer ball on the left or right side of the long bench? |

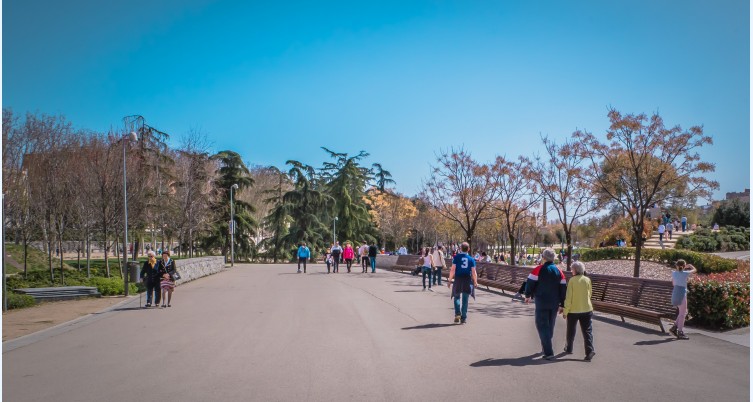

| **Options:** (A) right (B) left |
| --- |

| **Model** | **Reasoning & Output** |
| --- | --- |
| Qwen2.5-VL-7B-Instruct | The bench is located along the path. From this angle, the soccer ball appears to be positioned on the right side of the bench structure... 
 ✗ **Answer:** A |
| SAIL-VL2-8B-Thinking | *Locating the ball and the bench... The bench stretches along the right. The ball sits on the ground to the right of the bench's end... roughly aligned with the trees.* 
 ✗ **Answer:** A |
| VL-Rethinker-7B | *Analyzing spatial layout... The soccer ball is clearly visible on the right side relative to the bench... The perspective makes it appear further right.* 
 ✗ **Answer:** A |
| **SR-MCR-7B (Ours)** | *1. Analyze: Focus on the bench and surrounding elements. 2. Evaluate: (a) "right" does not indicate... (b) "left" suggests... 3. Conclusion: ...alignment with the bench's edge... indicates the ball's position relative to the bench.* 
 ✗ **Answer:** A |

