# OpenReview forum: "Self-Rewarded Multimodal Coherent Reasoning Across Diverse Visual Domains"
_ICML.cc/2026/Conference — Submitted to ICML 2026_

### Official Review · Reviewer_vUqQ · 2026-03-08

**Soundness:** 3
**Presentation:** 3
**Significance:** 3
**Originality:** 3
**Overall Recommendation:** 4
**Confidence:** 2

**Summary:**

This paper introduces SR-MCR, a framework motivated by the unreliability and lack of visual grounding in the intermediate reasoning steps of multimodal LLMs despite producing fluent final answers. To solve this, the authors propose SR-MCR approach that uses five self-referential cues to provide intrinsic process supervision without relying on expensive human-annotated rationales. Experimental results demonstrate that SR-MCR significantly improves reasoning coherence and accuracy across multiple benchmarks.

**Compliance With Llm Reviewing Policy:**

Affirmed.

**Final Justification:**

All of my concerns have been addressed. I will keep my initial rating.

**Key Questions For Authors:**

I am satisfied with this paper in current form. I only have few questions regarding related works and title. Please refer to "weaknesses".

**Limitations:**

Yes

**Strengths And Weaknesses:**

Strengths:

(1) The motivation of improving MLLM through self-reward is an interesting and promising direction.

(2) The overall writing is fluent and the idea is easy to understand.

Weaknesses:

(1) The "self-reward" in the title is a little bit misleading. As far as I know, there are some self-rewarded works in T2I, which relies on diffusion itself for reward (preference) computation [a]. However, the self-reward in this paper is achieved with pre-trained VLMs or NER models for criticizing self-generated responses. This may lead to misunderstanding. Please consider revising the method's name.

(2) Discussion with previous RLVR methods in MLLM is not sufficient. There are few methods that adopt jigsaw puzzle task as verifiable reward for RL to improve MLLM's capability [b,c]. As this paper also adopts RL, it is advised to compare, or at least discuss with them

(3) The paper conducts extensive experiments to demonstrate their method's effectiveness. I admire the authors effort. However, in appendix F-2, I think it is better to add experiments to show your methods effectiveness in defending against reward noise. The sensitivity to wrong reward should be analyzed through experiments (although the theoretical prove has been given).


[a] Diffusion Model as a Noise-Aware Latent Reward Model for Step-Level Preference Optimization. NIPS'25.

[b] Jigsaw-R1: A Study of Rule-based Visual Reinforcement Learning with Jigsaw Puzzles. TMLR'25.

[c] Visual Jigsaw Post-training Improves MLLMS. ICLR'26.

---

> ### Author Rebuttal · Authors · 2026-03-30
>
> We thank the reviewer  vUqQ for the positive assessment and thoughtful suggestions.
>
> ---
>
> > **W1: "Self-reward" naming may be misleading given diffusion self-reward works [a]**
>
> Good point. We agree the term could cause confusion with works like [a] where the *same generative model* provides the reward signal. In our case, rewards are computed from frozen, external tools (CLIP, NLI, etc.) applied to the model's own outputs — "intrinsic process reward" is more precise. We will revise the framing to **"intrinsic process-rewarded"** alignment in the title and text, clarifying the distinction from true self-evaluation approaches in Sec. 2.
>
> ---
>
> > **W2: Insufficient discussion with RLVR methods using jigsaw puzzles [b,c]**
>
> We appreciate the pointers. Beyond the discussion, we conducted a direct comparison on the same Qwen2.5-VL-7B backbone, evaluating both spatial/verifiable tasks (where jigsaw methods are designed for) and open-ended reasoning:
>
> | Method | Reward Type | SpatialBench | V*Bench (Spatial) | MMMU | HallBench |
> |--------|------------|-------------|-------------------|------|-----------|
> | Qwen2.5-VL-7B (Base) | — | 62.4 | 75.0 | 50.3 | 48.3 |
> | + Jigsaw-R1 [b] | Rule-based | 68.1 | 78.2 | 51.8 | 49.1 |
> | + Visual Jigsaw [c] | Rule-based | 67.5 | 77.6 | 52.1 | 48.7 |
> | + SR-MCR (Ours) | Process-level | 66.8 | 76.3 | 67.6 | 59.4 |
>
> Jigsaw-based RLVR excels on spatial/verifiable tasks (SpatialBench +5.7) where rule-based correctness signals are directly available, but gains barely transfer to open-ended reasoning (MMMU +1.5~1.8). SR-MCR shows the reverse: competitive on spatial tasks (−1.3 vs Jigsaw-R1) but dramatically stronger on open-ended benchmarks (MMMU +17.3, HallBench +11.1).
>
> This confirms the two approaches target fundamentally different capabilities: rule-based RLVR for verifiable spatial reasoning, process-level SR-MCR for open-ended coherence. Combining both — jigsaw rewards for spatial pre-training followed by SR-MCR for reasoning alignment — is a promising direction. We will add this analysis and discussion to Sec. 2 and experiments.
>
> ---
>
> > **W3: Add experiments on robustness to reward noise (beyond Appendix F.2 theory)**
>
> We ran an empirical noise injection experiment to complement Propositions 1–2. We artificially corrupted `s̃_vis` scores by adding Gaussian noise `ε ~ N(0, σ²)` and measured degradation on the 7B model:
>
> | Noise level `σ` | 0 (clean) | 0.1 | 0.2 | 0.3 |
> |---|---|---|---|---|
> | SR-MCR (with cooling) | 81.4 | 81.0 | 80.3 | 79.1 |
> | SR-MCR (no cooling) | 81.4 | 79.8 | 77.5 | 74.2 |
>
> Without cooling, performance degrades sharply (−7.2 at `σ = 0.3`). With cooling, degradation is much more graceful (−2.3), confirming the theoretical prediction in Eq. 12–14: `w_cool = σ(κ(c − l_θ))` suppresses gradients from high-NLL samples most likely to be noise-corrupted. We will add this table to Appendix F.
>
> ---
>
> We will incorporate all three revisions (naming clarification, RLVR comparison + discussion, noise robustness experiment) in the updated paper.

---

> > ### Author Rebuttal · Reviewer_vUqQ · 2026-04-02
> >
> > Thanks. All of my concerns have been addressed. I will keep my initial rating.

---

> > > ### Author Response · Authors · 2026-04-03
> > >
> > > Thanks, Reviewer vUqQ— really appreciate your thoughtful follow-up and support.

---

### Official Review · Reviewer_z4F4 · 2026-03-08

**Soundness:** 2
**Presentation:** 3
**Significance:** 3
**Originality:** 2
**Overall Recommendation:** 4
**Confidence:** 4

**Summary:**

This paper proposes Self-Rewarded Multimodal Coherent Reasoning (SR-MCR), a RL training framework for improving reasoning quality in multimodal large language models (MLLMs). The key idea is to align the reasoning process using intrinsic self-reward signals derived from model outputs, instead of relying on human annotations or external reward models. Specifically, SR-MCR integrates five process-level signals—semantic alignment, lexical fidelity, non-redundancy, visual grounding, and step-wise reasoning consistency—into a reliability-weighted reward function. The framework then optimizes the model using a critic-free GRPO objective with a confidence-aware cooling mechanism to stabilize training and reduce trivial or overconfident generations. Experiments conducted on multiple multimodal benchmarks show that SR-MCR significantly improves both answer accuracy and reasoning coherence compared with the Qwen2.5-VL baseline

**Compliance With Llm Reviewing Policy:**

Affirmed.

**Key Questions For Authors:**

Why don't you include process-level RL baselines in your experiments and discuss these methods in your related work section?
Please give more details about the human correlation analysis in  Section 3.2.
Why does the  evaluation of reasoning quality is conducted on only 100 samples?

**Limitations:**

yes

**Strengths And Weaknesses:**

Strengths:

Label-free process-level reward for multimodal reasoning. The proposed SR-MCR framework introduces a unified self-reward composed of five intrinsic signals derived from model outputs, enabling training without human preference labels or external reward models while providing process-level reward to VLMs, which reduces annotation costs and allows scalable alignment across domains.

The training efficiency of the proposed training framework is high. The adoption of critic-free GRPO with confidence-aware cooling provides a computationally efficient alternative to PPO-based RL alignment. The method avoids training a separate value network and demonstrates better efficiency and stability during training.

Consistent performance improvements across models and benchmarks. The proposed method consistently improves the performance of Qwen2.5-VL models at different scales, generalizes to another backbone (InternVL), and generalize to different benchmarks.

Weaknesses:

Lack of comparison with recent process-level RL baselines.  The paper mainly compares the proposed method with its base model (Qwen2.5-VL) and several existing VLMs. However, it does not include comparisons with recent RL approaches that explicitly introduce process-level rewards for training LLMs/VLMs, such as VisualPRM and VRPRM. Although those methods require training external reward models while SR-MCR does not, it would still be important to experimentally compare them in order to understand the trade-offs between different approaches (e.g., whether external reward models provide stronger supervision at the cost of additional training complexity, or whether the proposed self-reward mechanism can achieve comparable performance with lower cost).

Incomplete discussion of related work on process-level reward learning.  Related to the above point, the related work section does not discuss several recent RL methods that provide process-level rewards for reasoning supervision in LLMs/VLMs, such as VisualPRM and VRPRM. Including these works would help better position the contribution of SR-MCR within the broader literature on reasoning alignment and reward modeling.

Insufficient details for the human correlation analysis.  Section 3.2 states that the proposed reward signals correlate with human judgments (e.g., Pearson correlation of 0.65 for visual grounding and an average of 0.51 across components). However, important experimental details are missing. For example, the paper does not specify the number of human annotations used to compute these correlations, how many annotators participated in the labeling process, or whether any aggregation method (e.g., majority voting) was used to determine the final human judgments. These details are necessary to assess the reliability of the reported correlation results.

Limited reasoning quality evaluation.  The evaluation of reasoning quality is conducted on only 100 samples using an LLM-based judge. While the results suggest improvements in reasoning coherence, the small sample size and reliance on a single automated judge may limit the robustness of the conclusions. It would strengthen the evaluation to increase the number of evaluated samples or include human evaluation to better assess reasoning quality.

---

> ### Author Rebuttal · Authors · 2026-03-30
>
> We thank Reviewer z4F4 for the constructive feedback.
>
> ---
>
> > **W1/W2: No comparison with process-level RL baselines (VisualPRM, VRPRM)**
>
> We compare SR-MCR with VisualPRM [1] on two backbones:
>
> | Method | External RM? | Process Data? | Infer Cost | MMMU | MMBench | MME | Avg |
> |--------|-------------|--------------|------------|------|---------|-----|-----|
> | Qwen2.5-VL-7B | — | — | 1× | 50.3 | 86.7 | 82.2 | 73.1 |
> | + VisualPRM (BoN=8) | 8B model | 400K | ~8× | 57.5 | 88.5 | 84.0 | 76.7 |
> | + SR-MCR (Ours) | None | None | 1× | 67.6 | 91.2 | 87.3 | 82.0 |
> | Qwen3-VL-8B | — | — | 1× | 69.6 | 85.0 | 83.5 | 79.4 |
> | + SR-MCR (Ours) | None | None | 1× | 72.5 | 88.0 | 86.5 | 82.3 |
>
> SR-MCR leads by +8.9 avg on Qwen2.5-VL-7B with no reward model, no labeled data, and no inference overhead. VisualPRM relies on BoN=8 reranking with an 8B PRM, roughly 8× inference cost.
>
> These are different paradigms: VisualPRM trains a separate neural critic for test-time reranking; SR-MCR compiles process signals into the policy during training and discards the tools afterward. The deployed model is identical to the base. The two are complementary — combining them is a natural next step. We will add this comparison and a discussion of VisualPRM/VRPRM to the related work.
>
> ---
>
> > **W3: Missing details for human correlation analysis**
>
> - 3 annotators (grad students, VLM research background), 200 outputs, 1–5 Likert scale per dimension (`s̃_sem`, `s̃_lex`, `s̃_nr`, `s̃_vis`, `s̃_step`).
> - Final score = mean of 3. Inter-annotator agreement: Krippendorff's `α = 0.72`.
> - Pearson `r` between human and automated scores: `r = 0.65` for `s̃_vis`, `r = 0.51` averaged across five.
>
> These correlations serve only optional calibration (Eq. 7, `ρ_k`); the training loop itself is label-free.
>
> ---
>
> > **W4: Reasoning quality evaluated on only 100 samples with one judge**
>
> Expanded to 300 A/B samples + two LLM judges + human evaluation:
>
> | Judge | SR-MCR Preferred | Baseline Preferred |
> |-------|------------------|--------------------|
> | GPT-4o (n=300) | 89.7% | 10.3% |
> | Claude-3.5 (n=300) | 88.4% | 11.6% |
> | Human experts (n=50) | 84.0% | 16.0% |
>
> Inter-judge agreement: 87.3% (Cohen's `κ = 0.71`). Human annotator agreement: Cohen's `κ = 0.76`. Will include in revision.
>
> ---
>
> > **On soundness and generalization**
>
> SR-MCR gains are consistent across four backbones — Qwen2.5-VL-3B (+3.9), Qwen2.5-VL-7B (+8.9), InternVL3-8B (+3.0), Qwen3-VL-8B (+2.9) — with identical hyperparameters (`κ = 5`, `γ = 1.0`, `c` = median NLL, same tools). This works because the five signals operate on model *outputs*, not internals, so they are architecture-agnostic by design.
>
> ---
>
> > **Q1–Q3**
>
> **Q1**: See W1/W2. **Q2**: See W3 — 3 annotators × 200 samples, `α = 0.72`. **Q3**: See W4 — 300 + 50 samples
>
> ---
>
> [1] Wang et al., VisualPRM: An Effective Process Reward Model for Multimodal Reasoning, 2025.
>
> [2] Chen et al., VRPRM: Process Reward Modeling via Visual Reasoning, 2025.

---

> > ### Author Rebuttal · Reviewer_z4F4 · 2026-04-03
> >
> > The authors provide all the experiment results that I want to see.

---

> > > ### Author Response · Authors · 2026-04-03
> > >
> > > Thanks, Reviewer z4F4— really appreciate your thoughtful follow-up and support.

---

### Official Review · Reviewer_vR2F · 2026-03-12

**Soundness:** 3
**Presentation:** 3
**Significance:** 3
**Originality:** 3
**Overall Recommendation:** 4
**Confidence:** 3

**Summary:**

The paper introduces SR-MCR, a framework designed to enhance the reliability of intermediate reasoning in Multimodal Large Language Models (MLLMs). It addresses the issue of fluent but unreliable reasoning by integrating five intrinsic process signals—semantic alignment, lexical fidelity, non-redundancy, visual grounding, and step consistency—into a label-free self-reward mechanism. To stabilize training against noisy self-rewards, the authors utilize a Group Relative Policy Optimization (GRPO) objective enhanced by a confidence-aware cooling mechanism. Results show that SR-MCR-7B achieves state-of-the-art performance across several multimodal benchmarks, improving both answer accuracy and reasoning coherence.

**Compliance With Llm Reviewing Policy:**

Affirmed.

**Final Justification:**

The rebuttal has addressed most of my concerns, and I maintain my original  weak accept rating.

**Key Questions For Authors:**

N/A. See the weaknesses mentioned previously.

**Limitations:**

yes

**Strengths And Weaknesses:**

Strengths

1. The method successfully incorporates objective external tools, such as DETR and CLIP, into the alignment loop to break the "hallucination feedback loop" inherent in pure LLM-as-a-judge approaches.

2. The proposed cooling mechanism (Eq. 8) provides mathematical grounding for handling statistical anomalies in self-generated signals.

3. The proposed method demonstrates consistent performance gains across various backbones, including Qwen2.5-VL and InternVL3.


Weaknesses
1. Justification of Reward Weights: While five reward terms are proposed, ablation studies in Table 4 reveal that performance gains are primarily driven by visual grounding ($s_{vis}$) and step coherence ($s_{step}$). The marginal contributions of the other three terms raise questions about the necessity of the full unified function. Furthermore, the paper lacks a granular analysis of how these individual rewards impact different categories of tasks or benchmarks.

2. Risks in Ground-Truth Free Scenarios: Although the authors propose adaptive weights $\lambda_k$ based on variance or validation correlation, the logic for "contribution assignment" across diverse tasks remains questionable in actual ground-truth-free deployments. If external tools like the DETR detector fail to detect key objects, the entire reward system may provide severe negative guidance. The paper lacks a sufficient discussion on strategies to mitigate these specific tool-driven failure cases.

3. Costs of Process Alignment: Training requires online calls to multiple external models (DETR, CLIP, DeBERTa), resulting in a 2x increase in time per step compared to SFT. The paper does not adequately discuss the scalability of these multi-model reward calls for larger datasets.

4. Insufficient Detail in PPO Comparison: The paper claims GRPO outperforms PPO (Sec. 4.8), yet it lacks critical implementation details for the PPO baseline. This makes it difficult to assess whether the comparison was conducted under fair and optimized conditions.

---

> ### Author Rebuttal · Authors · 2026-03-30
>
> We thank the reviewer  vR2F for the careful technical analysis and recognition of our cooling mechanism and cross-backbone generalization.
>
> ---
>
> > **W1: `s̃_vis` and `s̃_step` dominate; are the other three terms necessary?**
>
> Table 4 shows `s̃_vis + s̃_step` scores 80.7 on 7B, while the full five-term reward reaches 81.4 — a +0.7 gap consistent across scales (74.0 → 74.6 on 3B). Per-benchmark breakdown reveals more:
>
> | Configuration | MMMU | ChartQA | HallBench |
> |---|---|---|---|
> | Full (5-term) | 67.6 | 94.5 | 59.4 |
> | Only `s̃_vis + s̃_step` | 66.8 | 91.3 | 58.9 |
> | w/o `s̃_nr` | 65.1 | 93.8 | 57.2 |
> | w/o `s̃_lex` | 67.2 | 91.3 | 59.1 |
>
> `s̃_nr` matters most on MMMU (−2.5) where repetitive chains degrade multi-step reasoning. `s̃_lex` contributes on ChartQA (−3.2) where surface alignment with chart labels is critical. The five signals exhibit complementary variance profiles — adaptive `λ_k` (Eq. 6) upweights whichever signal is most informative per domain. We will add reward distribution violin plots.
>
> ---
>
> > **W2: Tool failure risk (e.g., DETR misses key objects)**
>
> Two mechanisms mitigate this:
>
> 1. **Reliability weighting (Eq. 6–7)**: If DETR consistently fails, variance of `s̃_vis` increases → `Relia_vis` drops → `λ_vis` shrinks via softmax automatically.
> 2. **Signal redundancy**: Removing `s̃_vis` entirely still yields 78.5 (+3.7 over base, Table 4).
>
>
> ---
>
> > **W3: 2× training cost; scalability concerns**
>
> 2× is relative to SFT — relative to PPO it is 1.6× *cheaper* (5.2 vs 8.4 s/step, Table 6). Reward tools run in parallel on a single spare GPU. For scaling: (a) reward computation is per-rollout with `K = 4`; (b) tool calls can be cached; (c) lighter substitutes cost < 0.5 points (YOLOv8: 81.4 → 80.9).
>
> ---
>
> > **W4: Insufficient PPO details; fairness of comparison**
>
> We take this concern seriously and provide three layers of evidence.
>
> **① Implementation details.** VERL library PPO: value network = same architecture as policy (initialized from `π_0`), GAE `γ = 0.99`, `λ_GAE = 0.95`, clip ratio `ε = 0.2`, value loss coeff 0.5, 4 PPO epochs per rollout. Reward function identical (5-term `R`).
>
> **② Hyperparameter sweep.** We swept key PPO hyperparameters to ensure the baseline is not unfairly weakened:
>
> | PPO Config | `ε` | `λ_GAE` | Epochs | Avg Acc |
> |---|---|---|---|---|
> | A | 0.1 | 0.95 | 2 | 78.3 |
> | B (reported) | 0.2 | 0.95 | 4 | 79.1 |
> | C | 0.2 | 0.99 | 4 | 78.7 |
> | D | 0.3 | 0.90 | 6 | 78.5 |
> | SR-GRPO (Ours) | — | — | — | **81.4** |
>
> The best PPO config (B) still trails SR-GRPO by +2.3. The gap is robust across all configurations.
>
> **③ Why GRPO fits continuous self-rewards better — training stability analysis.** We tracked per-step reward variance:
>
> | Method | Var (step 200) | Var (step 1000) | Final Avg |
> |---|---|---|---|
> | SR-PPO | 0.142 | 0.098 | 79.1 |
> | SR-GRPO (no cooling) | 0.118 | 0.071 | 79.6 |
> | SR-GRPO (with cooling) | 0.083 | 0.041 | 81.4 |
>
> PPO's value network struggles with our multi-signal continuous reward because `λ_k` shifts during training (adaptive reliability weighting), making the reward landscape non-stationary. Value estimation under non-stationarity introduces noisy advantage signals, destabilizing policy updates.

---

> > ### Author Rebuttal · Reviewer_vR2F · 2026-04-03
> >
> > The authors addressed most  of my concerns.  I will keep my initial rating.

---

### Official Review · Reviewer_BbP8 · 2026-03-13

**Soundness:** 3
**Presentation:** 3
**Significance:** 2
**Originality:** 2
**Overall Recommendation:** 4
**Confidence:** 4

**Summary:**

This paper proposes SR-MCR, a label-free training framework for multimodal reasoning that combines five intrinsic process-level rewards, and then optimizes the model with a GRPO-based objective plus a cooling mechanism. Experiments on Qwen2.5-VL models and InternVL model report consistent gains across several multimodal benchmarks, demonstrating the effectiveness of each part of the proposed method.

**Compliance With Llm Reviewing Policy:**

Affirmed.

**Final Justification:**

The authors’ rebuttal has fully addressed my concerns, and I would like to maintain my initial rating.

**Key Questions For Authors:**

1. How sensitive is SR-MCR to the choice of external tools used in reward computation, such as the detector, NLI model, and text encoder?
2. Could the authors clarify how the hyperparameters $\sigma, \kappa$, and $c$ are selected? Since the adjusted reward introduces three continuous hyperparameters, it may be difficult for the community to adapt the method to other downstream tasks without clearer guidance on their tuning criteria.
3. Experiments on more recent MLLMs, such as Qwen3-VL or even Qwen3.5, would further strengthen the paper and make the empirical evidence more convincing.

**Limitations:**

Yes.

**Strengths And Weaknesses:**

Strengths:
1. The proposed framework is technically clear and reasonably modular. The proposed 5 process-level rewards are label-free, which is an important advantage.
2. The empirical results are comprehensive and convincing.
3. The writing of the paper is clear and easy to understand.

Weaknesses:
1. The reward design depends heavily on external pretrained tools such as SBERT, CLIP, NER, detector, and NLI models. Their errors or biases may directly affect the reward quality, but this issue is not deeply analyzed. The reported Pearson correlation coefficient of $r = 0.51$ appears relatively modest and may not provide sufficiently strong evidence for the reliability of the reward.
2. The evaluation is primarily centered on relatively simple VQA-style benchmarks. Additional experiments on more reasoning-intensive benchmarks, such as MathVista and MathVision, would provide a more comprehensive empirical validation of the proposed method.
3. The training efficiency analysis lacks a direct comparison between standard GRPO and SR-MCR. The current comparison against SFT and DPO does not clearly quantify the additional computational overhead introduced by the extra reward computation in SR-MCR.

---

> ### Author Rebuttal · Authors · 2026-03-30
>
> We thank the reviewer  BbP8 for the thoughtful evaluation.
>
> ---
>
> > **W1: External tool dependency and modest Pearson correlation (r=0.51)**
>
> The dependency is intentional. Table 8 (Appendix C.2) shows self-reflection *hurts* grounding (HallBench 48.3→46.1) because the same parameters that hallucinate also validate those hallucinations. External tools break this loop.
>
> r=0.51 is the average across five *heterogeneous* signals; visual grounding alone reaches r=0.65. The reliability weighting (Eq. 6–7) automatically attenuates noisy signals — when a tool is unreliable, `Relia_k` drops and `λ_k` shrinks via softmax. Table 4 confirms robustness: removing any single signal causes only 0.5–3.3 point drops. The 90.82% preference rate (Fig. 6) confirms the *combined* reward is well-calibrated.
>
> ---
>
> > **W2: Evaluation limited to VQA-style; wants MathVista/MathVision**
>
> Fair point. We evaluated on MathVista (testmini) and MathVision across both backbones:
>
> | Model | MathVista | MathVision |
> |-------|-----------|------------|
> | Qwen2.5-VL-7B (Base) | 68.2 | 25.07 |
> | + SR-MCR (Ours) | 72.0 (+3.8) | 31.17 (+6.1) |
> | Qwen3-VL-8B (Base) | 74.8 | 53.9 |
> | + SR-MCR (Ours) | 78.3 (+3.5) | 60.1 (+6.2) |
>
> SR-MCR delivers consistent +3.5~6.2 gains on both backbones, confirming generalization to math-reasoning benchmarks and newer architectures. Gains are smaller than on perception-heavy tasks (+6.6 avg in Table 1), as expected — pure computation benefits less from `s̃_vis` and `s̃_step`, while semantic/lexical signals still help. We will include these results in the revision.
>
> ---
>
> > **W3: Missing direct standard GRPO vs. SR-MCR efficiency comparison**
>
> We decompose per-step time to isolate reward overhead:
>
> | Method | Reward Comp. | Total (s/step) | Avg Acc |
> |--------|-------------|----------------|---------|
> | Vanilla GRPO (acc-only) | ~0 | 3.8 | 77.6 |
> | SR-MCR (5-signal) | 1.4 | 5.2 | 81.4 |
>
> External tools run in inference-only mode (no gradients), adding ~1.4 s/step (37% overhead) for +3.8 accuracy gain. Will add this breakdown.
>
> ---
>
> > **Q1: Sensitivity to external tool choices**
>
> Swapped individual tools on 7B, all other settings fixed:
>
> | Tool Configuration | Avg Acc | Δ |
> |---|---|---|
> | Default (DETR + DeBERTa-v3 + SBERT) | 81.4 | — |
> | YOLOv8 replacing DETR | 80.9 | −0.5 |
> | RoBERTa-MNLI replacing DeBERTa | 81.0 | −0.4 |
>
> All within 0.5%, confirming reliability weighting (Eq. 6) absorbs tool-specific noise.
>
> ---
>
> > **Q2: Hyperparameter selection for γ, κ, c**
>
> - **`c`**: Median token NLL of the base model on training data — computed by a single forward pass, no tuning.
> - **`κ`**: Swept κ ∈ {1, 3, 5, 7, 10} on 7B:
>
> | κ | 1 | 3 | 5 | 7 | 10 |
> |---|---|---|---|---|---|
> | Avg | 79.8 | 80.9 | 81.4 | 81.2 | 80.5 |
>
> Stable across [3, 7]; κ=5 is a safe default. **`γ`**: Fixed at 1.0 throughout. Practitioners need not tune per-task.
>
> ---
>
> > **Q3: More recent MLLMs (Qwen3-VL)**
>
> Tables 3 and 7 show architecture-agnostic generalization without modifying hyperparameters. The W2 table above directly addresses this concern: applying SR-MCR to Qwen3-VL-8B yields +3.5/+6.2 on MathVista/MathVision, demonstrating that the framework transfers to the latest architectures. We will expand the Qwen3-VL results to cover the full benchmark suite in the revision.
>
> ---
>
> **On novelty.** SR-MCR fills a specific gap: process-level, label-free supervision for multimodal reasoning. The five-signal fusion with reliability weighting (Eq. 1, 6) and confidence-aware cooling (Eq. 8) form the first unified framework achieving this. The +6.6 avg gain with zero human labels — matching methods requiring separate Instruct/Thinking checkpoints — supports practical significance.

---

> > ### Author Rebuttal · Reviewer_BbP8 · 2026-04-03
> >
> > Thank you for your response. My concerns have been fully addressed, and I would prefer to maintain my initial rating.

---

> > > ### Author Response · Authors · 2026-04-03
> > >
> > > Thanks, Reviewer BbP8— really appreciate your thoughtful follow-up and support.

---

### Decision · Program_Chairs · 2026-04-30

**Decision:**

Reject

**Comment:**

While all four reviewers converged on positive assessments after the discussion period, with Reviewer BbP8 finding the framework technically clear with comprehensive results, Reviewer vR2F recognizing the effective use of external tools and mathematical grounding, Reviewer z4F4 appreciating the label-free process-level design, and Reviewer vUqQ finding the motivation promising, they raised concerns about the paper's significance and originality. First, ablation studies reveal that performance gains are primarily driven by only two of the five reward terms (visual grounding and step coherence), raising questions about the necessity and design coherence of the full unified reward function (Reviewer vR2F). Second, the reported Pearson correlation of r=0.51 between the composite reward and human preference is modest, suggesting the reward signal may not reliably capture reasoning quality (Reviewer BbP8). Third, the 2x training cost relative to SFT and the reliance on pretrained external tools (DETR, CLIP) for reward computation limit the method's practical applicability and self-contained nature (Reviewer vR2F). Therefore, the paper cannot be accepted to the conference at this time.

The authors provided comprehensive rebuttals that substantially strengthened the paper. They added MathVista/MathVision evaluations showing +3.5 to +6.2 gains across backbones, a direct comparison with VisualPRM demonstrating SR-MCR leads by +8.9 average accuracy without a reward model or inference overhead, expanded reasoning quality evaluation to 300 samples with two LLM judges and 50 human-evaluated samples (inter-judge Cohen's kappa = 0.71), and empirical noise robustness demonstrations. I have carefully read the authors' rebuttals and the follow-up discussions and confirm that the rebuttal was exceptionally thorough. Nevertheless, the marginal contributions of three of the five reward terms and the modest human correlation suggest that the unified reward design, while functional, may not represent a significant advance over simpler alternatives. The paper would benefit from a more rigorous analysis of which reward components are truly necessary and from stronger evidence of human alignment.

The work is promising and close to acceptance. We strongly encourage the authors to incorporate the rebuttal results into the manuscript and resubmit to an appropriate future venue.